# The tumor microenvironment shows a hierarchy of cell-cell interactions dominated by fibroblasts

Shimrit Mayer [1,3], Tomer Milo [2,3], Achinoam Isaacson[1], Coral Halperin[1], Shoval Miyara [2], Yaniv Stein[1], Chen Lior[1], Meirav Pevsner-Fischer[1], Eldad Tzahor [2], Avi Mayo[2], Uri Alon[2] ✉ & Ruth Scherz-Shouval [1] ✉

The tumor microenvironment (TME) is comprised of non-malignant cells that interact with each other and with cancer cells, critically impacting cancer biology. The TME is complex, and understanding it requires simplifying approaches. Here we provide an experimental-mathematical approach to decompose the TME into small circuits of interacting cell types. We find, using female breast cancer single-cell-RNA-sequencing data, a hierarchical network of interactions, with cancer-associated fibroblasts (CAFs) at the top secreting factors primarily to tumor-associated macrophages (TAMs). This network is composed of repeating circuit motifs. We isolate the strongest two-cell circuit motif by culturing fibroblasts and macrophages in-vitro, and analyze their dynamics and transcriptomes. This isolated circuit recapitulates the hierarchy of in-vivo interactions, and enables testing the effect of ligand-receptor interactions on cell dynamics and function, as we demonstrate by identifying a mediator of CAF-TAM interactions - RARRES2, and its receptor CMKLR1. Thus, the complexity of the TME may be simplified by identifying small circuits, facilitating the development of strategies to modulate the TME.

Tumors are ecosystems in which cancer cells and diverse non-malignant cells of the tumor microenvironment (TME) interact with each other. These interactions impact all aspects of cancer biology including tumor progression, metastasis and response to treatment[1–3]. Approaches to understand and modulate the TME are thus major goals of cancer biology.

The TME is complex and heterogeneous. It is composed of cancer associated fibroblasts (CAFs), tumor associated macrophages (TAMs), T cells, NK cells, B cells, endothelial cells, pericytes and other cell types. The non-malignant cells of the TME are genomically stable but plastic in the sense that their transcriptomes and phenotypes are sculpted by interactions with cancer cells and other cells of the TME. Fibroblasts, for example, are rewired into diverse myofibroblastic, immune-regulatory and antigen-presenting CAFs. These subpopulations have distinct functions related to ECM production, adhesion and immune

regulation[4–6]. Bone-marrow derived monocytes differentiate into macrophages that can acquire pro- or anti-inflammatory states in response to different signals including cytokines and hypoxia[7–9], contributing to phenotypic plasticity of diseased tissues[10,11].

The complexity of the TME poses significant challenges for analysis and modulation. One can consider two extreme possibilities: If all cell types strongly interact with all other cells, such that the network is non-decomposable to smaller parts, understanding this network of interactions may be difficult. At the other extreme, the network is composed of a small set of recurring circuits, each of which has autonomy in the sense that its dynamics and behavior are preserved when the circuit is isolated. In this case, an understanding of the entire network can be built up by analyzing each small circuit separately. Notably, the latter possibility is found in intracellular networks such as gene regulation networks (GRNs) which display recurring and

[1]Department of Biomolecular Sciences, The Weizmann Institute of Science, Rehovot, Israel. [2]Department of Molecular Cell Biology, The Weizmann Institute of Science, Rehovot, Israel. [3]These authors contributed equally: Shimrit Mayer, Tomer Milo. ✉e-mail: uri.alon@weizmann.ac.il; ruth.shouval@weizmann.ac.il

autonomous network motifs[12,13]. These are recurring patterns of regulation, such as coherent and incoherent feedforward loops, that occur in many different systems. Each motif has biological functions which can be studied in isolation, such that the system's behavior can be built up from its basic motifs[12,13].

Here we ask whether such network-motif structure and circuit autonomy might occur between cells in the TME. We use single-cell RNA-sequencing (scRNA-seq) data to define the network of interactions between cell types in human breast cancer TMEs. Our analysis reveals a hierarchical network structure, with CAFs at the top of the hierarchy. To further explore the network dynamics, we use network-motif analysis and identify a recurring two-cell circuit motif. The strongest instance of this motif is a circuit containing CAFs and TAMs. Within this circuit, CAFs can support themselves by means of an autocrine loop and engage in mutual paracrine interactions with TAMs. We then study the dynamics and functions of this circuit by culturing fibroblasts and macrophages in-vitro, and find that it has bistability with a viable steady state in which both cell types proliferate and die, maintaining a fixed ratio. The interaction strengths and gene expression profiles of the in-vitro circuit recapitulate those of the in-vivo circuit, and enable testing the effect of ligand-receptor interactions on cell dynamics and function, as we show by identifying RARRES2, a potential mediator of CAF-TAM interactions, and its receptor CMKLR1. Thus, the complexity of the TME may be amenable to reductionist analysis by identifying and isolating small cell circuits.

## Results

### Analysis of the breast cancer microenvironment reveals a hierarchy of interactions with a dominant CAF-TAM circuit

To begin to untangle the complexity of the TME, we mapped the network of cell-cell interactions by analyzing published scRNA-seq data from breast cancer patients[14]. We identified the cell types (Fig. 1a and S1a) and scored their interactions (Supplementary Data 1) using CellChat[15], a tool for estimating ligand-receptor interaction strength. We found multiple pairs of interacting cell types (Fig. 1b), and used CellChat to score the strength of each interaction.

We found that the strongest interaction occurred between CAFs and myeloid cells (comprised mainly of tumor-associated macrophages, hereafter referred to as TAMs), followed by the interaction between CAFs and mast cells (Fig. 1b), and an autocrine loop in which CAFs send ligands which they also sense by expressing their receptors. In fact, CAFs were the cell type with the highest interaction scores. These features are found also in mouse breast cancer scRNA-seq data[16]. In mice, as in human data, CAFs are the most interacting cell type and their autocrine interaction, as well as their interaction with macrophages, are among the strongest (Fig. 1b, Supplementary Fig. 1b). In normal breast tissue (from healthy individuals, see methods), the strongest interaction of fibroblasts is also with myeloid cells (comprised mainly of macrophages, hereafter termed macrophages), however both macrophages and fibroblasts engage in strong interactions with other cell types (Supplementary Fig. 1c, Supplementary Data 1).

To better understand the structure of the interactions among the cell types, we constructed a weighted and directed cell-network from the interaction matrix (Fig. 1c, Supplementary Fig. 1d; See Methods for details). This analysis revealed that the interaction network is hierarchical and mostly feedforward, with CAFs at the top sending signals to the other cell types, and TAMs at the bottom, receiving signals from CAFs and other cell types (Fig. 1c).

We next asked whether this complex network of interactions can be simplified and described in terms of repeating instances of smaller circuits, which we can isolate and further explore. For this purpose we employed network motif analysis, which detects small circuits that occur in the network significantly more often than in randomized networks. We began with circuits made of two interacting cell types. Of

the seven possible connected two-cell circuits (Fig. 1d), we found that only one type recurs with an interaction score that exceeds those found in randomized networks, and is thus a network motif[12,13]. This circuit has two interacting cell types that send mutual paracrine signals, and each also has an autocrine signaling loop. This circuit appears three times in the network, and all have CAFs as one of the nodes. Of these circuits the strongest circuit in terms of total and mean interaction score is the CAF-TAM circuit (Fig. 1e). We also analyzed circuits made of three and four cell types (Fig. 1f, g, Supplementary Fig. 1e). The most common circuits were made of combinations of the above-mentioned two-cell circuit (Fig. 1e). The CAF-TAM pair participated in the highest scoring instances of these three- and four-cell circuits (Fig. 1g, Supplementary Fig. 1e). For example, the CAF-TAM circuit interacts with cancer cells to form a three-cell circuit in which CAFs send signals to cancer cells which in turn send signals to TAMs (Fig. 1h).

We also constructed a weighted and directed cell-network from the interaction matrix of the normal breast tissue. This analysis showed that fibroblasts are still near the top of the hierarchy and macrophages are at the bottom, however the network shows no significant 2-node motifs (Supplementary Fig. 1f, g).

We conclude that the breast cancer TME interaction network is hierarchical and composed of repeated occurrences of a specific two-cell circuit motif. CAFs are at the top of the hierarchy and send out many signals, whereas TAMs are at the bottom receiving end. The CAF-TAM two-cell circuit is an example of this motif and is one of the circuits with the strongest interactions in the network. Thus, studying this circuit in isolation can improve our understanding of the TME interaction network.

### The isolated macrophage-fibroblast circuit shows bistability with a viable steady state

The discovery of a cell circuit motif in the TME raises the question of its dynamics and autonomy. If one isolates the circuit and permits the cells to interact without the remaining TME context, do the cell populations reach a viable steady state, and do they recapitulate some of their TME functions?

To test this, we analyzed the fibroblast-macrophage circuit. We co-cultured fibroblasts from the mammary fat pad together with bone marrow derived macrophages (BMDMs) from syngeneic BALB/c mice and tracked cell populations over time. We tracked cell growth from different initial concentrations of fibroblasts and BMDMs by flow cytometry after 3 and 7 days of co-culture (Supplementary Fig. 2a), and displayed their dynamics in a *phase portrait*[17]. The phase portrait is a geometric representation of the dynamic behavior of a system. It has two axes: the fibroblast (X-axis) and macrophage (Y-axis) cell counts. Arrows (vectors) on the phase portrait indicate how cell counts change from day 3 to day 7 (Fig. 2a). The phase portrait uses many initial conditions and two timepoints to infer the dynamics from any initial condition at all timepoints, by following the arrows.

On the Y-axis are macrophages in mono-culture, without fibroblasts. These cells could not promote their own growth (Fig. 2b). However, co-culture with fibroblasts supported macrophage growth, and revealed dynamic interactions (Fig. 2c). At sufficiently high initial concentrations, both cell types reached a stable steady state - a point to which arrows converge from all directions (green dot, Fig. 2c). In this state, macrophages and fibroblasts continually turn over as indicated by EdU incorporation measurements (Supplementary Fig. 2b, c; see Methods), but maintain their numbers in a dynamic steady state. We designate this state the ON state. In the context of wound healing, such a state of mutually supporting fibroblasts and macrophages is termed 'hot fibrosis'[9,17,18].

Below a threshold combination of concentrations, both cell types decline to zero (red dot, Fig. 2c). This is another possible steady state, called the OFF state. This state is the expected outcome of successful resolution of acute injury or acute inflammation.

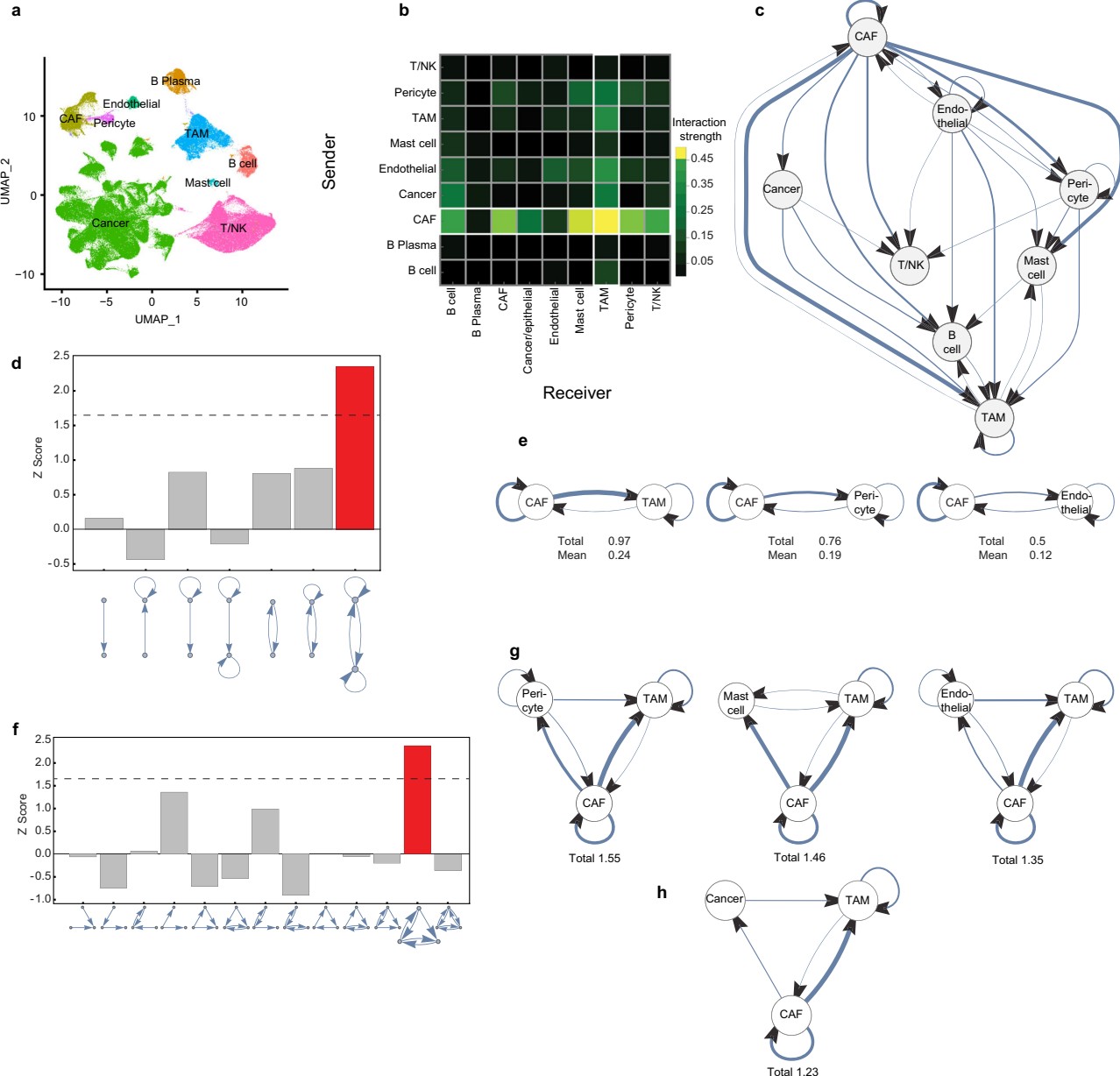

**Fig. 1 | Network analysis reveals a hierarchy of interactions with a dominant CAF-TAM circuit in the breast tumor microenvironment. a** UMAP visualization of the main cell clusters in the breast TME in human scRNA-seq data from 32 patients[14]. **b** Heatmap of interaction strengths between pairs of cells based on cumulative ligand-receptor interaction scores using CellChat[15] applied to the scRNA-seq data of (**a**). **c** Illustration of the structure of the network based on the analysis in (**b**) shows hierarchy (the root node was chosen as the node with the highest weighted outdegree). Arrow width is proportional to the interaction score.

**d** Network motif analysis of all 7 possible two-cell circuit patterns, tested for abundance relative to randomized degree-preserving networks (bootstrapping $n = 10,000$, see Methods). The dashed line represents a 0.05 $p$-value threshold. **e** The top three two-cell interaction subgraphs scored by the average weight all include CAFs. **f** Network motif analysis of the 13 possible three-cell-circuit patterns. The dashed line represents a 0.05 $p$-value threshold. **g** The top three scoring three-cell circuit subgraphs. **h** The three-cell circuit of CAFs, TAMs and cancer cells. Source data are provided as a Source Data file.

When fibroblasts grow alone, as seen on the X-axis of the phase portrait, their dynamics depend on their initial cell numbers. Below a critical threshold, which is an unstable fixed point (white dot, Fig. 2c), fibroblast numbers decline to zero. Above this threshold, fibroblasts are able to maintain themselves, and their numbers rise to a fixed point called the ON-OFF state (fibroblasts are ON, macrophages are OFF, half-yellow dot, Fig. 2c). Fibroblasts at this fixed point continually turn over in a dynamic steady-state, as indicated by EdU incorporation (Supplementary Fig. 2b).

In physiological terms, a state in which fibroblasts maintain high numbers in the absence of macrophages is referred to as 'cold

fibrosis'[18–20], and it is distinct from the ON state, which has macrophages together with fibroblasts.

We tested the robustness of the phase portrait assay in several ways (Supplementary Fig. 2d–i). Biological replicates of the experiment gave rise to similar phase portraits (Supplementary Fig. 2d). Phase portraits derived from cell counts at days 7, 14, and 21 showed qualitatively similar dynamics as the cell counts derived from days 3 and 7 (Supplementary Fig. 2e), suggesting approximate temporal invariance. We repeated the analysis using a different mouse strain, C57BL/6, and obtained a similar phase portrait (Supplementary Fig. 2f). To accurately measure growth dynamics at very low initial cell

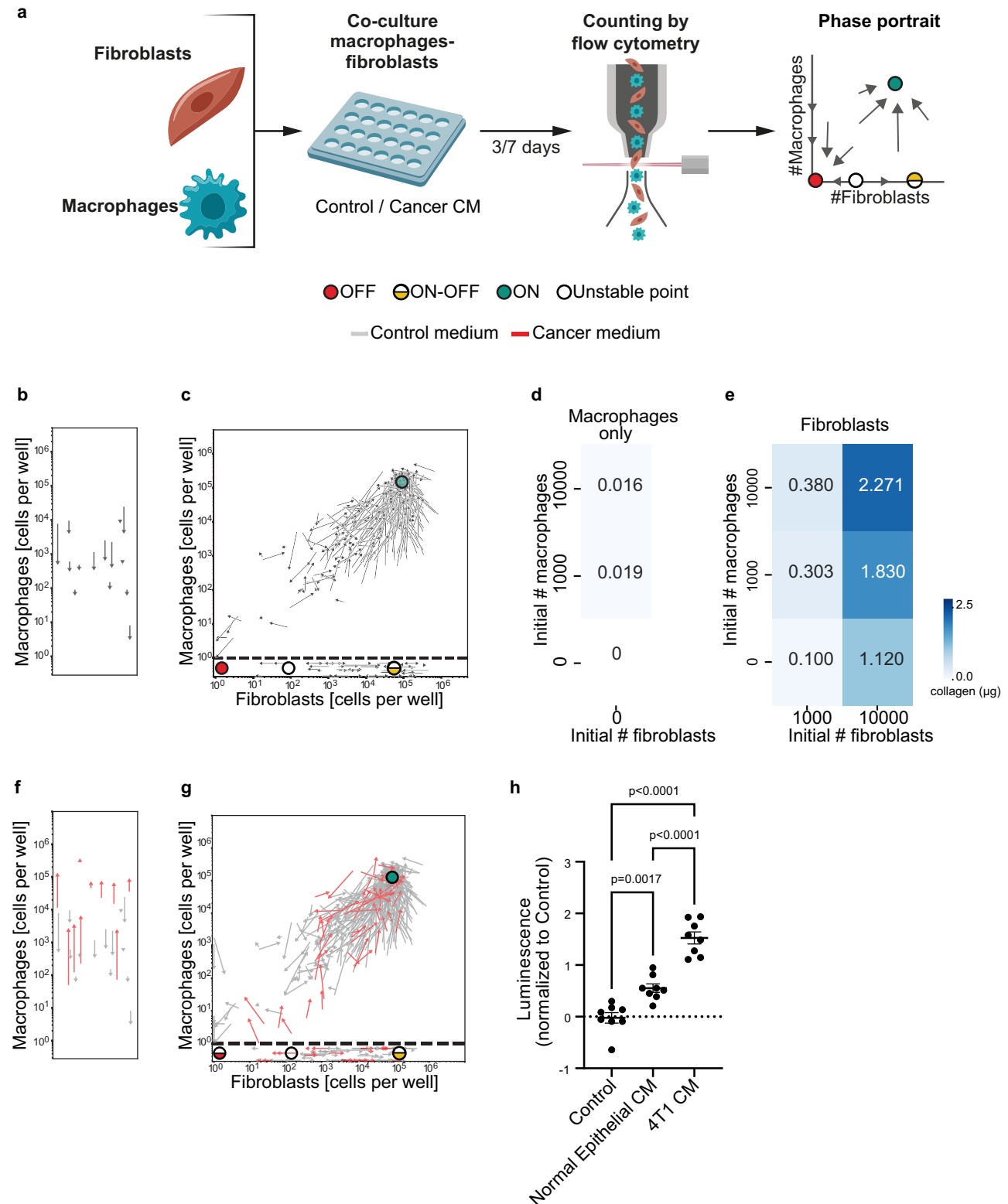

concentrations (several cells per well) we scaled the experiment up from 96-well plates to 6-well plates, which have a 30-fold larger area. We accordingly multiplied cell numbers measured in 96-wells by a factor of 30. We found qualitative agreement in the overlapping regions of effective concentrations in 96 and 6 well assays (Supplementary Fig. 2g, red vs. gray arrows). We also tested co-cultures of BMDMs with fibroblasts from two other organs—lung and mesometrial

fat. The resulting phase portraits were qualitatively similar, with the same fixed-point structure (Supplementary Fig. 2h, i). We conclude that the phase portrait assay is robust and replicable under multiple different conditions.

We further asked whether addition of macrophages to an on-going culture of fibroblasts—simulating the infiltration of BMDMs into a tissue populated by resident fibroblasts—would yield similar

**Fig. 2 | Isolation of the macrophage-fibroblast circuit in-vitro allows analysis of its dynamics and reveals bistability with a viable steady-state at high cell numbers. a** Illustration of the experimental procedure. Macrophages and fibroblasts were isolated from mice, co-cultured at different ratios for 3 or 7 days in control or cancer CM, and counted by flow cytometry. **b**, **c** Experimental phase portraits of macrophage-fibroblast dynamics in-vitro. Arrow tails represent cell counts at day 3 of co-culture, and arrowheads represent cell counts at day 7 (starting from the same initial cell concentration). **b** Mono-cultured macrophages. **c** Fibroblasts co-cultured with macrophages are presented above the horizontal dashed line, mono-cultured fibroblasts are below the line. Fixed points are denoted by dots: "ON": green; "OFF": red; unstable: white; "ON-OFF" state: half-yellow. The following number of biologically independent samples was used: macrophages only: $n = 5$; macrophages with fibroblasts: $n = 24$. The positions of the fixed points were determined by the modeling of Fig. 3. **d**, **e** Quantification of the amount of fibrillar collagen deposited after 7 days of co-culture of macrophages and fibroblasts. Macrophages only: $n = 4$; fibroblasts and fibroblast-macrophage co-cultures:

$n = 3$ biologically independent samples. Data are presented as mean. **f** Experimental phase portrait of macrophages grown in mono-culture in the presence of 4T1 cancer CM (red arrows), overlayed on the control phase portrait presented in (**b**) (gray arrows; performed in parallel to control media cultures); $n = 10$ biologically independent samples. **g** Experimental phase portrait of macrophage-fibroblast dynamics following in-vitro co-culture with 4T1 cancer CM (red arrows), overlayed on the control phase portrait presented in (**c**) (gray arrows). Fibroblasts co-cultured with macrophages are presented above the horizontal dashed line; mono-cultured fibroblasts are presented below the line (performed in parallel to control media co-cultures); $n = 12$ biologically independent samples for the cancer CM. **h** Macrophage cell numbers following three days of growth in mono-culture in the presence of Control or CM from normal mouse epithelial cells or from 4T1 cancer cells; with $n = 8$ biologically independent samples. *P*-value was calculated using one way ANOVA. Error bars represent ± SEM. **b–h** All data are combined from at least three independent experiments. Source data are provided as a Source Data file.

interaction dynamics compared to those observed by simultaneous plating of both cell types. We observed similar convergence towards the ON state when macrophages were either added to the cultures 3 days after the initial plating of fibroblasts, or simultaneously plated with fibroblasts, suggesting that the interaction dynamics are independent of this variable (Supplementary Fig. 2g, pink arrows). This finding further supports the conclusion that the ON-OFF state is semi-stable—arrows converge to it on the x axis (changes to fibroblast numbers) but point away from it along the y direction (changes to macrophage numbers).

To evaluate not only growth but also function, we monitored ECM deposition—a key fibroblast function, known to be supported by macrophages, in the co-culture setup. We assessed ECM deposition by measuring fibrillar collagen levels in different regions of the phase portrait (Fig. 2d, e, see Methods). As expected, macrophages alone did not deposit collagen, whereas fibroblasts in monoculture did (Fig. 2e). Co-culture with macrophages resulted in a 2–3 fold increase in fibroblast collagen deposition, and maximal collagen deposition was measured near the ON state, suggesting that this state is not only the joint steady-state of the two cell types, but also the state of highest ECM production (Fig. 2e). These findings support the notion that the fibroblast-macrophage circuit in co-culture maintains growth as well as functionality.

The CAF-TAM circuit occurs in vivo together with cancer cells that significantly affect the TME[21,22]. To evaluate the impact of cancer cells we compared growth in control medium to breast cancer conditioned medium (CM). We obtained the cancer CM from 4T1 triple-negative breast cancer cells syngeneic to the fibroblasts and BMDMs grown for 48 h. We added the cancer CM to co-cultures of mammary fibroblasts and BMDMs, and obtained their phase portrait (Fig. 2f, g; red arrows). In the presence of cancer CM (Fig. 2f, g) the phase portrait behaved differently than that of the control medium (Fig. 2c). Macrophages in cancer CM were able to grow in the absence of fibroblasts (red arrows; Fig. 2f), in contrast to control media, in which their growth depends on fibroblasts (gray arrows; Fig. 2f). This may relate to the composition of 4T1-CM which contains factors that regulate macrophage proliferation[23]. A growth-promoting effect was also observed when macrophages were grown in CM from normal (non-malignant) mammary epithelial cells, however this effect was mild compared to the growth-promoting effect of cancer CM (Fig. 2h).

The ability of macrophages to grow without fibroblasts led to a change in the OFF state. This state, which was stable in the control medium, became semi-stable in the presence of cancer CM, and was lost when macrophages were added. The ON and ON-OFF states are still observed with cancer CM (Fig. 2g).

The phase portraits highlight the dynamic nature of the fibroblast-macrophage interactions, the codependence of macrophages and fibroblasts, and the effect of cancer CM on these dynamics.

## Mathematical modeling infers growth interactions and dynamics of the fibroblast-macrophage circuit

The experimental phase portraits provide a global view of the dynamic behavior of the cell populations. To elucidate the forces that govern these dynamics in different growth conditions, we developed a mathematical model of interacting fibroblasts (F) and macrophages (M) (Fig. 3a, b, equations provided in Methods). Our model simplifies a more complex model of biochemical reactions[17,24], in order to provide a minimal number of effective interaction parameters. This simplification makes it possible to infer the parameters from the data without identifiability or overfitting concerns using a simple regression approach (see Methods).

The model has 4 parameters per cell type (Fig. 3b). Fibroblasts are removed at rate $r_F$. Their proliferation is induced by paracrine interactions from macrophages at rate $p_{MF}$, and by an autocrine loop at rate $p_{FF}$. Fibroblast numbers cannot exceed a carrying capacity $K_F$ - the maximal cell population that prohibits further growth - which is determined by environmental factors such as nutrients and space availability[17,25]. Four analogous parameters define macrophage dynamics: removal $r_M$, paracrine and autocrine interactions $p_{FM}$ and $p_{MM}$, and a carrying capacity $K_M$.

We estimated the values of the model parameters by fitting cell numbers at day 7 given their numbers at day 3 (Fig. 3c–e, see Methods). Many distributed different initial conditions allowed us to infer the dynamics for any initial condition at all timepoints. The model showed good fits for the experimental dynamics, explaining 84%–93% of the variance in the data (Supplementary Fig. 3a–d) with a reasonably low error in predicting the direction of growth of the cell populations (Supplementary Fig. 3e, f). We also validated convergence and robustness of parameter calibration by the Bayesian tool, PyDREAM[26] (Supplementary Fig. 4a–c).

The inferred circuit models give rise to theoretical phase portraits (Fig. 3f, g). These phase portraits are similar to the experimental ones, and help to fill out regions that were difficult to reach experimentally (e.g., low cell numbers). The phase portraits show the ON, ON-OFF, OFF-ON, and OFF fixed points, as well as the unstable fixed points. All of these fixed points can be calculated based on the inferred parameters (Fig. 3c–e, Supplementary Fig. 3g, h and Supplementary Table 1). The inferred phase portraits also delineate the basins of attraction (i.e., the regions in which trajectories converge to a given fixed point) of the ON and OFF states in the control medium, and of the ON and OFF-ON states in the cancer CM (shaded in different colors; Fig. 3f, g). Additionally, the model provides inferred carrying capacities (K), which are about 10-fold greater for macrophages than for fibroblasts, consistent with previous findings[17,25] (Supplementary Fig. 3i, j).

The mathematical model also allows us to plot the rate of change in cell numbers for each cell type on the phase portrait (Fig. 3h–k). In this 'heatmap' plot, lack of dependence on fibroblast numbers is

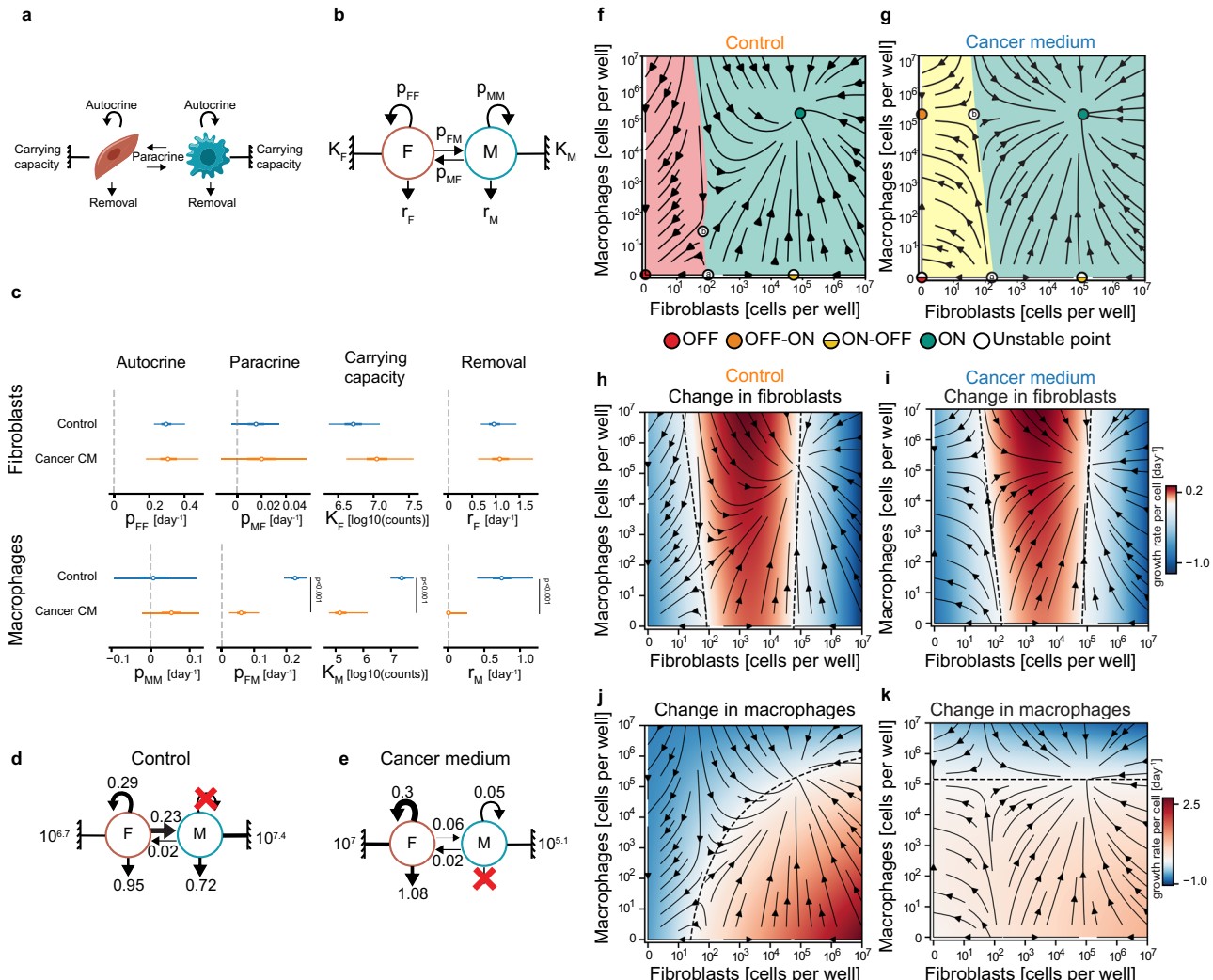

**Fig. 3 | Circuit interactions and dynamics inferred by mathematical modeling.**
**a**, **b** Theoretical cell circuits with the model parameters: $p_{FF}$ - fibroblast autocrine loop; $K_F$ - fibroblast carrying capacity; $r_F$ - fibroblast removal rate; and $p_{MF}$ - paracrine effect of macrophages on fibroblasts. Analogous parameters for the macrophages are: removal $r_M$; paracrine and autocrine interactions $p_{FM}$ and $p_{MM}$, respectively; and carrying capacity $K_M$. **c** Best fit values for the parameters of the fibroblast-macrophage circuits in control and cancer CM. The distribution of each parameter is presented by its median (circle), interquartile range (thick line), and 95% confidence interval (CI; thin line). *P*-values and CI were calculated by bootstrapping (*n* = 5000) and contrast distributions (see Methods). **d**, **e** Theoretical cell circuits with the mean value of each parameter for control and cancer CM. Inferred

phase portraits showing the population dynamics of macrophages and fibroblasts in control (**f**) or cancer CM (**g**). Basins of attraction are indicated by color: in the control medium cells can flow to the "OFF" state (red dot) if they start in the red region or to the "ON" state (green dot) if they start in the green region. A population of fibroblasts that resides to the right of the unstable fixed point, denoted a, will flow to the "ON-OFF" state (half-yellow dot). In the cancer CM the flow in the red region changes (indicated with a yellow region) - it drains to the "OFF-ON" state (orange dot). **h**–**k** Heatmaps indicating the predicted average growth rate of fibroblasts and macrophages in control and cancer CM. Red indicates growth and blue indicates shrinkage of the cell population. Dashed lines are the nullclines of the system; along them there is no change in the cell population.

characterized by fixed horizontal shades, whereas lack of dependence on macrophage numbers is characterized by fixed vertical shades.

The theoretical phase portraits (Fig. 3f, g) and their associated heatmaps (Fig. 3h–k) help visualize the interdependence of fibroblasts and macrophages, and the dramatic effect of cancer CM on the circuit dynamics. In the control medium, fibroblasts barely depend on macrophages, whereas macrophages are heavily influenced by fibroblasts. This is also evident by the inferred circuit parameters (Fig. 3c, d), in which fibroblasts support their own growth through a strong autocrine loop and support macrophages through a strong paracrine interaction. Macrophages signal back with a much weaker paracrine interaction and have a very weak autocrine loop.

Cancer CM changes the circuit dynamics. The phase portrait contains a new stable OFF-ON state of macrophages (Fibroblasts are OFF, macrophages are ON; Fig. 3g, orange dot), a shift of the unstable fixed point *b* to a higher macrophage concentration, and a change in

the OFF state from stable in the control medium to semi-stable in cancer CM. Although fibroblasts below a critical concentration still flow to the OFF state, addition of macrophages causes the cells to flow to the new OFF-ON state with macrophages alone (Fig. 3g, orange dot drains the yellow region). Fibroblast growth rate is therefore self-sustaining regardless of their seeding ratio with the macrophages. Their effect on macrophages is reduced but non-zero, because the measured macrophage growth rate still increases with fibroblast number in CM.

Notably, in cancer CM, the proliferation rates and inferred circuit interactions align with the hierarchy of the network from the in-vivo scRNA-seq data (depicted in Fig. 1c). Fibroblasts remain highly dependent on themselves due to a strong autocrine loop and are minimally affected by macrophages (Fig. 3i). Additionally, the hierarchy of the network indicates that macrophages are the main recipients of signals from different cells, which suggests a decreased

dependence on fibroblast growth factor secretion and a reduction in growth factor paracrine interactions (Fig. 3k). The new OFF-ON fixed point in which macrophages lose their dependence on fibroblasts is explained by the inferred circuit in cancer CM (Fig. 3e). The removal rate of macrophages is zero, signifying enhanced survival in the CM. Thus, cancer CM changes all macrophage parameters to allow their enhanced growth.

We also calculated models for the two other organs from which we produced fibroblasts—lung and mesometrial fat. The model showed global similarity of the phase portraits in the three organ contexts. Nevertheless, their inferred circuits were somewhat different from the mammary circuit (Supplementary Fig. 3k–n). Mechanistically, this may suggest that the ON state is achieved differently in different organs. It also highlights the added value of a mathematical approach when comparing different organs or disease contexts.

We conclude that the dynamics in the co-culture are generated by an inferred circuit of interactions which is very similar to the circuit determined from analysis of scRNA-seq of the human breast TME. The hierarchy of interaction strengths is recapitulated by the in-vitro circuit, with the fibroblast autocrine and paracrine interactions stronger than those of the macrophages. Macrophages are thus more dependent on external growth conditions - be it reciprocal signaling with fibroblasts or factors secreted to the medium by cancer cells, whereas fibroblasts are more self-sufficient and can support their own growth.

### The isolated circuit recapitulates in-vivo transcriptomic profiles

In addition to the dynamics of the circuit, we asked whether the molecular phenotypes of the cells in the isolated co-culture circuit recapitulate the in-vivo molecular phenotypes. We therefore performed RNA-seq of fibroblasts and macrophages from co-cultures at concentrations near the ON, ON-OFF, and OFF-ON states, in control and cancer CM (Supplementary Fig. 5). Under normal growth conditions (control medium), co-culture with fibroblasts strongly affected the macrophage transcriptome, as indicated by clustering analysis (Fig. 4a, first split). Fibroblasts were not affected as strongly by co-culture with BMDMs (Fig. 4b), supporting the interaction hierarchy by which macrophages are affected by fibroblasts more than fibroblasts by macrophages.

To characterize these expression changes we performed pathway analysis using Metascape[27] (Supplementary Data 2–3). Macrophages in mono-culture expressed genes involved in cell cycle and DNA related pathways. These were attenuated in the presence of fibroblasts and greatly reduced with fibroblasts and cancer CM. Instead, these conditions led to upregulation of migration, chemotaxis and inflammation regulation (Fig. 4c). Fibroblast transcriptomes were generally unaffected by the presence of macrophages. In contrast, cancer CM modulated their transcriptome, though it did not alter it to the extent that it did to macrophages. Cancer CM led to upregulation of vascular development genes, ECM organization and support of epithelial growth, but maintained many other pathways found in the control medium (Fig. 4d).

To further understand whether the state of the cells in co-culture mimics the in-vivo state, we measured the enrichment of TAM and CAF signatures in the co-culture RNA-seq data, using single sample gene set enrichment analysis (ssGSEA)[28] with a protumorigenic signature for TAMs[29], and iCAF, myCAF and apCAF signatures for CAFs (Supplementary Data 4)[30]. Macrophages co-cultured with fibroblasts or in cancer CM showed enrichment of a pro-tumorigenic TAM signature, and this enrichment was significantly induced by co-culture with both fibroblasts and cancer CM (Fig. 4e, Supplementary Data 4).

We further confirmed the protumorigenic shift in macrophages at the protein level, by examining the cell-surface expression levels of CD206, a known protumoral marker of TAMs[7]. Co-culture with cancer CM led to upregulation of CD206 on macrophages, as expected[10]. However the expression of CD206 was significantly induced by the

addition of fibroblast CM to the cancer CM, further demonstrating the key role of fibroblasts in rewiring macrophages towards a protumoral phenotype (Fig. 4f).

CAFs are heterogeneous and comprise of diverse populations with distinct tasks—myofibroblastic CAFs (myCAF) that harbor ECM and wound healing regulatory modules, inflammatory/immune regulatory CAFs (iCAF) characterized by a secretory phenotype and immune cell regulatory activity, and antigen presenting CAFs (apCAF) expressing MHC class II molecules[8]. To test whether CM induces CAF-like transcriptional signatures, we applied ssGSEA[28] on fibroblasts grown in control or cancer CM. ssGSEA analysis of fibroblasts grown in control medium demonstrated enrichment for both myCAF and iCAF signatures. (The apCAF signature was not detected in normal fibroblasts, consistent with our previously published notion that apCAFs are most likely not derived from tissue resident fibroblasts[6]). In the presence of cancer CM, however, the iCAF signature was no longer detected and the myCAF signature dominated the population (Fig. 4g). Macrophage co-culture had little effect on these scores.

In summary, the genes, pathways and signatures revealed by the transcriptomic analysis, and their similarity to those found in mouse and human tumors, highlight the potential value of a combined experimental and mathematical cell-circuit approach to better understand cell-cell interactions in the TME.

### The isolated circuit highlights RARRES2 and CMKLR1 as potential mediators of CAF-TAM signaling

The isolated cell circuit enables testing the effect of specific ligand-receptor interactions on the dynamics, function and transcriptomes of the circuit. We therefore asked which ligands are upregulated in both human and mouse CAFs, as well as in the co-culture. First, we identified shared ligands between human and mouse breast CAFs based on scRNA-seq datasets[14,16], using the NicheNet tool[31]. This analysis revealed 77 shared ligands which have potential receptors on TAMs (Fig. 5a). Next, we assessed the expression of these ligands in the RNA-seq data from our co-culture (Fig. 5b). We identified a cluster of 23 ligands upregulated in the presence of cancer CM (Fig. 5b, cluster 1). We validated the expression of several genes from this cluster by qRT-PCR and found that indeed they were significantly upregulated in fibroblasts co-cultured in the presence of cancer CM (Fig. 5c, Supplementary Fig. 6c).

This cluster included genes well known to mediate CAF-TAM interactions in cancer, such as *Tgfb1*, *Csf1*, and *Ccl2*[32]. It also included Retinoic Acid Receptor Responder (*Rarres2*; also known as Chemerin), a chemokine known to regulate inflammation, adipogenesis, and metabolism through activation of the chemokine-like receptor 1 (CMKLR1)[33,34]. *RARRES2* was recently found to be expressed by CAFs in colorectal cancer[35]. However, its role in the TME was not elucidated. We therefore decided to focus our analyses on RARRES2. We found that *Rarres2* is upregulated in fibroblasts cultured in the presence of cancer CM, and its expression is further induced by co-culture with macrophages (Fig. 5c). Strikingly, its receptor, *Cmklr1*, is upregulated in macrophages upon co-culture with fibroblasts or in the presence of cancer CM (Fig. 5d). To directly test whether RARRES2 is not only expressed by CAFs but also secreted by them, we isolated CAFs from 4T1 tumors, cultured them for 3 days and measured the levels of RARRES2 in the medium by ELISA. We also isolated and cultured the 4T1 cancer cells themselves, as control (Fig. 5e). This analysis confirmed that RARRES2 is secreted from CAFs and not from cancer cells. To gain insight into how RARRES2 affects TAMs, we monitored macrophage proliferation and migration in the presence of control medium, cancer CM, and cancer CM with recombinant RARRES2. RARRES2 did not affect macrophage proliferation (Supplementary Fig. 6d). It did however significantly enhance macrophage migration (Fig. 5f), highlighting the potential role of RARRES2-CMKLR1 signaling in breast cancer.

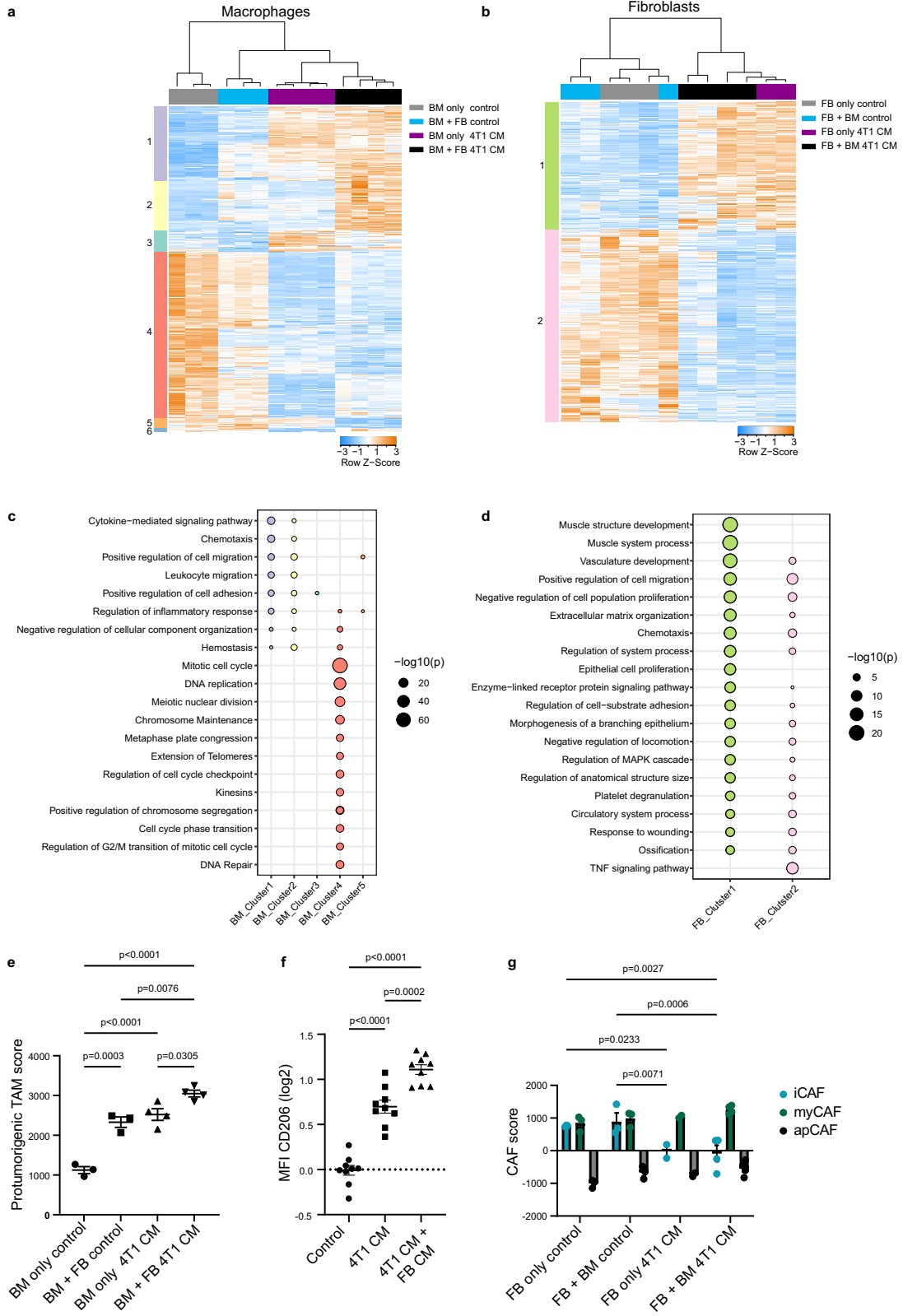

To test the relevance of our findings to human disease, we compared *RARRES2* expression in CAFs to fibroblasts from normal breast. Analysis of the human breast scRNA-seq dataset[14] showed that *RARRES2* is specifically upregulated by CAFs and not by normal mammary fibroblasts (Fig. 5g). Moreover, it is uniquely expressed by CAFs and not by other cell types within the human breast TME (Fig. 5h, Supplementary Fig. 6a). Next, to explore the potential effects of

*RARRES2* expression on TAMs in patients, we computed the correlation between CAFs expressing high levels of *RARRES2* and macrophages expressing a protumorigenic TAM signature associated with poor survival[29] in the human breast cancer RNA-seq dataset[14]. We found a strong correlation, consistent with *RARRES2* expression contributing to a protumoral TAM phenotype (Fig. 5i). To further assess the clinical relevance of *RARRES2* expression, we compared the expression levels

**Fig. 4 | RNA sequencing supports predicted changes in macrophage and fibroblast cell circuits in cancer-conditioned medium. a, b** Heatmaps showing hierarchical clustering of differentially expressed genes (DEGs; basemean >5; |LogFoldChange | > 1; FDR < 0.1). **a** An interaction model (medium and culture) was used to compare DEGs between macrophages mono-cultured (only) or co-cultured, with mammary fibroblasts in DMEM vs cancer CM. The mono-cultured macrophages in DMEM were collected at day 0 (since they cannot maintain themselves in DMEM for 7 days), and at day 7 in cancer CM. The co-cultured macrophages were collected after 7 days of co-culture with mammary fibroblasts, in either DMEM or cancer CM. Macrophages in DMEM: $n = 3$ biologically independent samples, macrophages in cancer CM: $n = 4$ biologically independent samples. **b** An interaction model (medium and culture) was used to compare DEGs between fibroblasts mono-cultured (only), or co-cultured, with macrophages in cancer CM vs. DMEM. Fibroblasts in DMEM: $n = 3$ biologically independent samples for each condition, Fibroblasts in cancer CM: only $n = 2$ biologically independent samples, co-cultured $n = 4$ biologically independent samples. **c, d** Pathway analysis of the macrophage clusters from (**a**) and fibroblast clusters from (**b**) was conducted using Metascape[27].

Selected significant pathways are shown, see full list in Supplementary Data 2–3 (FDR < 0.05). **e** The ssGSEA score of the protumorigenic TAM signature was applied to the macrophage RNA-seq data. Same biological replicates as indicated in (**a**). **f** Flow cytometry analysis was conducted on macrophages to evaluate the expression of the protumorigenic marker CD206. The macrophages were stained after being mono-cultured in the presence of control medium, 4T1-cancer CM, or a 1:1 mix of cancer CM and CM from fibroblasts induced by cancer (see Methods). $n = 9$ biologically independent samples from a total of three separate experiments, each experiment was normalized to the control mean fluorescence intensity (MFI). **e, f** $P$-value was calculated using one-way ANOVA followed by Tukey's multiple comparisons test. Error bars represent ± SEM. Same biological replicates as indicated in (**a**). **g** The ssGSEA scores of the iCAF, myCAF and apCAF signatures were applied to the fibroblast RNA-seq data. $P$-value was calculated using two-way ANOVA followed by Tukey's multiple comparisons test. Error bars represent ± SEM. Same biological replicates as indicated in (**b**). Source data are provided as a Source Data file.

of *RARRES2* in low-grade *vs* high-grade tumors. We found that *RARRES2* expression is significantly higher in high-grade cases compared to low-grade cases (Fig. 5j). We conclude that RARRES2, identified through our in-vitro circuit approach, mediates CAF signaling to TAMs in mouse models of breast cancer and in human disease, and may serve as a therapeutic target for future exploration.

## Discussion

In this study we present an approach to analyze the TME by breaking it down into smaller cell circuits. We find that the network of cell interactions in the breast TME is hierarchical with CAFs at the top. This network consists of recurring instances of a two-cell circuit motif. The strongest instance of this circuit is CAFs with an autocrine loop and paracrine signal to TAMs, which have weaker paracrine and autocrine signals. We tested the autonomy of this two-cell circuit by growing fibroblasts and macrophages in co-culture in control and cancer-CM. The in-vitro circuit recapitulates the hierarchy of the in-vivo interaction strengths. It shows bistability with a viable steady-state in which the two cell types support each other. It also recapitulates much of the transcriptomic phenotype seen in-vivo. The in-vitro circuit allows identifying molecular players and testing the effects of modulating them, as we demonstrate by identifying a potential mediator of CAF-TAM interactions - RARRES2, and its receptor CMKLR1. We show that this mediator enhances macrophage migration in vitro and is correlated with protumorigenic TAM phenotypes and high tumor grade in breast cancer patients. Thus, the TME may in principle be broken down into small cell circuits that can be profitably studied in isolation using co-culture.

To discover recurring circuits we employed a network motif approach. In the past, this approach was primarily used to analyze intra-cellular networks such as gene regulatory networks (GRN[12,13,36]). It revealed recurring regulatory circuits within cells, such as feedforward loops, and formed the basis for understanding the logic of large gene circuits by breaking them down into understandable smaller circuits[12,13]. Here we applied network motif analysis to the network of interactions between cells, rather than within a cell. We studied cell-cell interactions in the breast TME with the aim of simplifying its analysis in a similar way to GRNs. We first determined the network using ligand-receptor analysis from scRNA-seq. The network has a hierarchical structure, with CAFs at the top sending out the majority of signals, and TAMs at the bottom, receiving signals from other cells.

Within this network we find a dominant two-cell circuit motif, in which two cell types have autocrine loops and paracrine mutual signaling. The strongest instances have CAFs as one of the nodes. We also analyzed motifs of 3 and 4 cell types, and found that the higher order motifs are comprised of combinations of this two-cell circuit motif. This suggests that the two-cell circuit motif may be an elementary

building block of the TME. The network for normal tissue showed no significant 2-node motifs. Motif analysis may thus help to discover important differences between normal and disease states, by revealing cell circuits that are crucial in each case.

In order to be useful as a basic building block, however, a circuit must have a degree of autonomy, in the sense that it preserves its dynamic and biological function even when isolated from the rest of the network. Such autonomy is biologically useful, in analogy to modules in a machine—it ensures that the circuit works reliably no matter what the state of the rest of the network is[37,38]. From the point of view of research, autonomy is crucial for a reductionist approach, as it posits that isolating parts of a system can contribute to the understanding of its whole.

We find that the fibroblast-macrophage circuit is indeed autonomous—it recapitulates the interactions and transcriptomes when grown in culture in-vitro, especially in the presence of cancer CM. The circuit has the dynamic feature of bistability, in which depending on initial cell concentrations, it can reach one of two stable steady-state fixed points. One of these states is a viable steady-state in which both cells turn over and mutually support each other. The other steady state has zero cell types. Such bistability was recently exploited therapeutically in the context of abrogating fibrosis in mice models in the heart[39] and liver[40].

The complete dynamical mapping of the circuit in co-culture, known as a phase portrait, allows one to infer the interaction strengths between the cells using a mathematical model approach. We find that the inferred interaction strengths largely recapitulate the in-vivo strengths in the cell-cell network, although they are obtained in a completely independent way. The strongest interactions are the fibroblast autocrine and paracrine interactions, whereas the macrophage interactions are much weaker. Similarly, the effect of co-culture on macrophage dynamics is much stronger than the effect of co-culture on fibroblasts.

The circuit co-culture allows dissecting the effect of cancer by means of growth in cancer CM. Cancer CM preserves the bistability property and the viable two-cell steady state. This is an aspect of the circuit's dynamical autonomy. However it destabilizes the steady-state of zero cells, and instead creates a new fixed point with macrophages on their own. Such a macrophage-only state, in which macrophages turn over and support their own growth, may resemble aspects of macrophage activation syndrome and autoinflammation[41–43]. Just as in the in-vivo network, in which cancer cells send more signals to TAMs than to CAFs, cancer CM seems to affect macrophages much more than fibroblasts. This may imply that specific cancer cell signals are critical to shape the dynamics, despite the fact that cancer cells are located in the middle of the hierarchy in the in-vivo network.

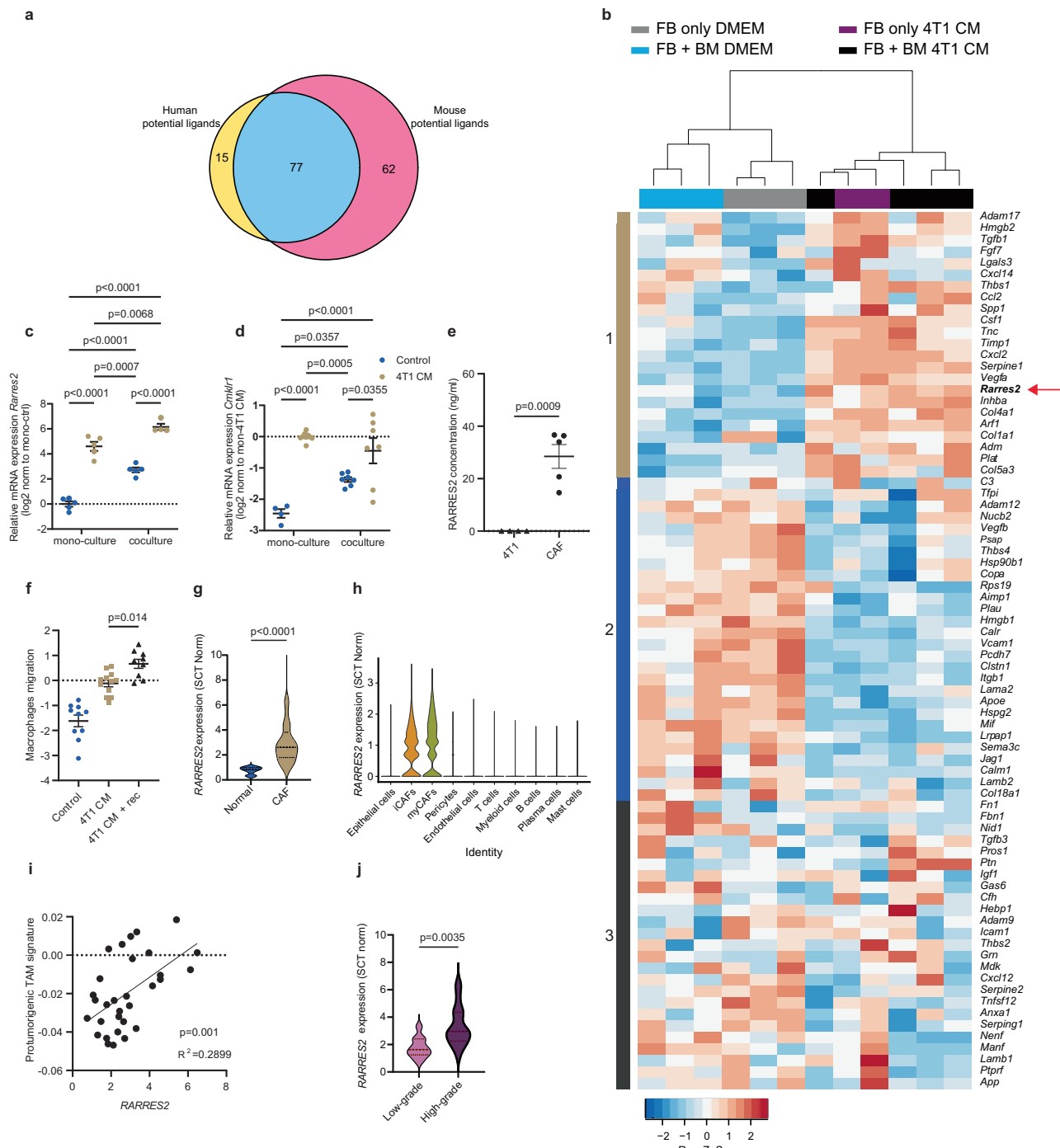

**Fig. 5 | RARRES2 and its receptor CMKLR1 are part of the CAF-TAM signaling axis in breast cancer. a** Venn diagram of potential CAF-to-TAM ligands shared between breast cancer patient tumors[14] and 4T1 tumors in mice[16] based on the NicheNet ligand-receptor tool. **b** Heatmap representation of the expression of the 77 shared ligands from (**a**) in fibroblasts from the in vitro cell-circuit, based on bulk RNA-seq data from Fig. 4b. **c, d** qPCR of *Rarres2* and *Cmklr1* from fibroblasts and macrophages. mono-cultured or co-cultured in control or cancer CM for 72 h. Data are combined from at least three independent experiments. *P*-value was calculated using two-way ANOVA followed by Tukey's multiple comparisons test. Error bars represent ±SEM. **c** Fibroblasts mono-cultured in control or in cancer CM *n* = 5, fibroblasts co-cultured in control *n* = 5 or in cancer CM *n* = 4 biologically independent samples. **d** Macrophages mono-cultured in control or in cancer CM *n* = 4 or in cancer CM *n* = 9, macrophages co-cultured in control or in cancer CM *n* = 7 biologically independent samples. **e** RARRES2 secretion levels in the media were assessed by ELISA from CAF *n* = 5 and 4T1 *n* = 4 biological biologically independent samples. *P*-value was calculated using two-sided students' *t* test, Error bars represent ± SEM.

**f** Transwell migration assay of macrophages in the presence of control medium *n* = 10, or 4T1 cancer CM with *n* = 9 or without *n* = 12 recombinant RARRES2 for 24 h, n indicates biologically independent samples. Luminescence values were normalized to 4T1-CM in log2. Data are combined from at least three independent experiments. P-value was calculated using one-way ANOVA followed by Tukey's multiple comparisons test. Error bars represent ± SEM. **g** Violin plot of *RARRES2* expression based on human scRNA-seq of normal fibroblasts *n* = 17 versus CAFs *n* = 32. *P*-value was calculated using two-sided students' *t* test. **h** *RARRES2* expression in different clusters from human scRNA-seq data, including CAF subclusters (defined using markers shown in Supplementary Fig. 6a). **i** Linear regression between *RARRES2* expression in fibroblasts and TAM signature in macrophages. Each dot represents one patient, based on human scRNA-seq[14]. *P*-value was calculated using *F*-test for linear regression. **j** *RARRES2* gene expression in breast cancer patients[14] stratified by grade, high grade = grade 3, *n* = 21; low-grade = grade 1 and 2, *n* = 10 patients. *P*-value was calculated using two-sided students' *t* test. Source data are provided as a Source Data file.

Gene expression analysis confirms that macrophages are more affected by context - presence of fibroblasts or cancer CM - than fibroblasts. Co-culture with cancer CM shifts their gene expression from self-maintaining, cell-cycle and DNA replication genes to protumorigenic TAM genes. Fibroblasts are pushed by cancer CM towards a myCAF phenotype with ECM deposition as the characteristic upregulated pathway.

Having established the autonomy of this two-cell circuit, we asked whether it may help to identify molecular players in the TME and to characterize the effects of modulating such factors. By intersecting transcriptional patterns found in tumors and in our isolated co-culture system, we identified the chemokine RARRES2 as a paracrine signal sent from fibroblasts to macrophages and from CAFs to TAMs, inducing CMKLR1 upregulation and macrophage migration. Previous studies have indicated that RARRES2 plays a significant role in mediating cell trafficking to sites of inflammation and demonstrates angiogenic properties[44,45]. In various cancers, RARRES2 can function as both a pro- and anti-inflammatory mediator, depending on the context[35,46]. However, its role in the breast TME and specifically in breast CAFs remains largely unexplored. We find that in breast cancer patients, RARRES2 is expressed specifically in CAFs - it is not expressed in normal fibroblasts or in any other cell type in the breast TME. Importantly, its expression increases in high-grade tumors and its levels are correlated with a protumorigenic TAM signature, highlighting the relevance of our findings to human disease, and the potential applicability of the circuit approach to help identify novel players and understand their biological role.

One limitation of our study is the use of cancer CM, which may not fully replicate the complexity of interactions observed. For instance, the experimental setup used to derive the cancer CM lacked fibroblasts and macrophages, and hence lacked the feedback from these cells to the cancer cells. Another limitation is the lack of spatial effects in the analysis. Recent work in the context of kidney fibrosis has demonstrated how local inflammation and hypoxia fields can affect a circuit's parameters, and thus the same circuit may have different steady-states in different parts of the tissue[9]. Emerging spatial omic approaches can help to reveal the effects of such spatial fields on the circuits, and to form a quantitative understanding of the heterogeneous tumor landscape.

It would be important in future work to test the impact of modulating the other molecular factors identified in this study that underlie the circuit interactions, such as factors involved in the fibroblast autocrine loop. Combinatorial targeting of these factors will likely be required to provide therapeutic benefit. One could also extend this study by adding cell types to understand the three- and four-cell circuits. We envision a research program to understand the small circuits and to build from them a complete understanding of the TME with its multiple cell types. Such a quantitative circuit understanding can guide rational approaches to modulating the TME for cancer therapy.

## Methods

### Ethics statement
All animal studies were conducted in accordance with the regulations formulated by the Institutional Animal Care and Use Committee (IACUC; protocol #05420621-2). BALB/c and C57BL/6 mice were purchased from Harlan Laboratories and maintained under specific-pathogen-free conditions at the Weizmann Institute of Science (WIS) animal facility. The light-dark cycle was 12 h. The ambient temperature was 22 Celsius degrees and humidity was between 35 and 55%.

### Cancer cells
4T1 female murine triple negative breast cancer cells were a generous gift from the lab of Zvika Granot (HUJI, Israel, originally from ATCC). These cells were transduced to express green fluorescent protein (GFP) using the FUW-GFP vector. 4T1-GFP cells were cultured in

Dulbecco's modified Eagle's medium (DMEM; Biological Industries, 01-052-1 A) with 10% fetal bovine serum (FBS; Invitrogen) and 5% P/S (Biological Industries). Cell lines were tested routinely for Mycoplasma using EZ-PCR Mycoplasma Test Kit (#20-700-20, Biological Industries). cell lines were maintained below passage 10.

### 4T1 condition medium
4T1 cells were seeded at $1 \times 10^6$ cells/ml in 10 cm plates. 24 h later (when the cells have formed a monolayer) the medium was replaced with fresh medium. 72 h later, the medium was collected, filtered through a 0.22 μm strainer, and diluted with DMEM with 20% FBS, at a ratio of 1:1.

### Normal mammary fat pad and mesometrial fat fibroblasts isolation
Normal mammary fat pad and mesometrial fat fibroblasts were isolated and dissociated from the mammary fat pads or the fat tissue of two (BALB/c or C57BL/6, 8 weeks old) females per each biological replicate. organs were minced and dissociated using a gentle MACS dissociator, in the presence of an enzymatic digestion solution containing 1 mg/ml collagenase II (Merck Millipore, 234155), 1 mg/ml collagenase IV (Merck Millipore, C4-22) and 70 U/ml DNase (Invitrogen, 18047019), in DMEM. The samples were filtered through a 70 μm cell strainer into cold PBS, and cells were pelleted by centrifugation at 350 g for 5 min at 4 °C, and resuspended in red blood cell lysis buffer (BioLegend 420301), then washed with PBS and centrifuged at 350 g for 5 min at 4 °C. Mammary and fat fibroblasts were seeded on collagen I (Sigma-Aldrich, Cat. C3867) coated 10 cm or 6-well plates, respectively. The cells were expanded for 6 days in DMEM with 5% P/S and 10% of FBS and the media was replaced every 3 days.

### Primary lung fibroblast isolation
Lungs of BALB/c female (8 weeks old) were excised, dissociated, minced, and incubated with enzymatic digestion solution containing 3 mg/ml collagenase A (Sigma Aldrich, 11088793001) and 70 unit/ml DNase in RPMI 1640 (Biological industries, 01-100-1 A) using a gentle-MACS dissociator, 30 min at 37 °C. The samples were filtered through a 70-μm cell strainer into cold PBS and cells were pelleted by centrifugation at 350 g for 5 min at 4 °C and resuspended in red blood cell lysis buffer, then washed with PBS and pelleted at 350 g for 5 min at 4 °C. Lung fibroblasts were seeded onto 10 cm plates coated with collagen I. The cells were expanded for 5 days in DMEM with 5% P/S and 10% FBS, and medium was replaced after 3 days.

### Macrophage differentiation
Bone marrow-derived macrophages from BALB/c (8 weeks old) female mice were differentiated into macrophages by growth in DMEM with 10% FBS, 5% P/S and 20% L929 CM on a petri dish. The medium was replenished at day 3, and the macrophages were reseeded for the experiment on day 7.

### Macrophage-fibroblast co-culture
The fibroblasts and the macrophages were isolated and expanded separately for 7 days, after which the fibroblasts were trypsinized and resuspended in an ice-cold MACS buffer (PBS with 0.5% BSA). The samples were pelleted by centrifugation at 350 g for 5 min at 4 °C, incubated with anti-EpCAM (Miltenyi, 130-105-958) and anti-CD45 (Miltenyi, 130-052-301) magnetic beads, transferred to LS columns (Miltenyi, 130-042-401), and the fibroblast-enriched, CD45/EpCAM-depleted, flow-through was collected. The macrophages were harvested with non-enzymatic cell dissociation solution (Biological Industries,03-071-1B) and washed with PBS without calcium and magnesium (PBS (-/-)). The macrophages were stained with 2 μM CFSE and seeded together with the fibroblasts in 96-well or 6-well plates pre-coated with collagen I. The co-cultures were grown in DMEM with 10%

FBS and 5% P/S, or with 4T1-CM (performed in parallel to control media co-cultures). Every 3 days 50 µl/1 ml of medium (for 96 well/6 well, respectively) were replaced with fresh medium. Macrophages and fibroblasts were seeded at different concentration ranges ($0–10^5$ in 96 well and $0–5 \times 10^6$ in 6 well), with the same combination of cell concentrations seeded in parallel onto two different plates. Plates were analyzed by Flow cytometry, one at day 3 and the other at day 7. Cell counts from 96-well plates were multiplied by 30 to scale for 6-well plates. We chose day 3 as the initial time point (and not an earlier time point such as day 0), to ensure the cells are settled in the 2D layer in terms of secreted factor interactions.

During the first day after seeding interactions occur in 3D within the entire well volume, since it takes time for the cells to settle and form a 2D layer at the bottom. Thus, their effective density during this phase is very low and below the separatrix, predicting that their numbers should crash. Indeed, we seeded cells with varied initial conditions spanning the entire range of the phase plane and observed that cell numbers at day 3 are much smaller than at seeding.

### Flow cytometry for cell quantification

Fibroblasts and macrophages were harvested from tissue culture plates by incubation with a non-enzymatic cell dissociation solution, washed, and transferred to round-bottom 96-well plates. The cells were then counted by flow cytometry using CFSE and anti-CD11b-Pacific blue antibody (Biolegend, Cat.101224) as positive markers for macrophages. Cells stained negatively for these markers were counted as fibroblasts. Dead cells were excluded using DRAQ7 (Biolegend, Cat. 424001). Flow cytometry was performed using CytoFlex-S (Beckman Colter). FACS analysis was performed using Flowjo software v.10.7.1.

### Flow cytometry for CD206 marker

A total of $1 \times 10^5$ BMDM were seeded in collagen pre-coated 96-well plate and cultured for 72 h in the presence of different mediums: Control medium: 100 ul of DMEM with 10% FBS. 4T1 cancer CM: 100 ul of 4T1 cancer CM (as described above). 4T1 cancer CM + FB CM: A mixture of 100 ul of cancer CM and 100 ul of fibroblast CM induced by 4T1 CM for 48 h. Then, macrophages were harvested from tissue culture plates by incubation with a non-enzymatic cell dissociation solution, washed, and transferred to round-bottom 96-well plates. Cells stained for anti-CD11b-Pacific blue antibody (Biolegend, Cat.101224) and anti-CD206-BV711 (Biolegendi, Cat.141727).Dead cells were excluded using PI (Sigma Aldrich, P4170). Flow cytometry was performed using CytoFlex-S (Beckman Colter). FACS analysis was performed using Flowjo software v.10.7.1.

### EdU (5-ethynyl-20 -deoxyuridine) assay

Mammary fibroblasts and macrophages were co-cultured in 96-well plates at a range of concentrations ($0,1 \times 10^3, 1 \times 10^4$ and $3 \times 10^4$), and an EdU incorporation assay was performed on day 7. EdU (10 mM) was added to the cells for 2 h, after which the cells were harvested, stained with the live/dead exclusion marker Ghost-Dye-Violet450 (TONBO, Cat.13-0863), and with anti-CD45-FITC (Miltenyi Biotec, Cat.130-110-658). EdU incorporation was detected using the Click-iT Plus EdU Flow Cytometry Assay Kit according to the manufacturer's instructions (ThermoFisher, Cat. C10634). Samples were then acquired using a CytoFlex-S (Beckman Colter), macrophages were gated based on positive staining for CD45, and fibroblasts were called based on negative staining for this marker. Analysis was performed with FlowJo 10.7.1.

### Collagen deposition measurement in-vitro

Fibroblasts and macrophages were seeded in mono-culture or co-culture ($0, 1 \times 10^3$ and $1 \times 10^4$ cells), in 200 ul of DMEM, in collagen I pre-coated 96-well plates. Per experiment, at least two technical replicates per condition were used. Cells were left for 7 days in culture to assure

confluence before performing collagen content measurement using a commercial Sirius Red collagen staining kit (Chondrex, Cat.9046), and measured by Cytation 5-Imaging Reader (Biotek). The collagen measurements obtained are normalized to the total protein amount per well.

### Cell size determination

Fibroblasts and macrophages were seeded in mono-culture at $3 \times 10^4$ cells in 8-well slide-containing chambers (Ibidi, Cat.80826) that were pre-coated with collagen I. After 7 days, the cells were fixed in 4% paraformaldehyde (PFA) for 10 min at RT, washed twice with PBS (-/-), and stained with DAPI (to mark nuclei), and CellMask™ Deep Red plasma membrane stain (ThermoFisher, Cat.C10046), according to the manufacturer's protocol. Images were taken with a Nikon Eclipse Ci microscope ×10 objective. Segmentation was done using Cellpose[47] with a Flow threshold of 0.8 and a cell probability threshold of −1 on the DAPI and CellMask channel. The cells that touched the borders were removed, and the cell sizes were quantified by QuPath[48] using the Cellpose segmentation.

### Bulk RNA-seq

We performed RNA-sequencing of the co-cultures at the ON state. As control, we analyzed mono-cultured fibroblasts and macrophages. Fibroblasts from different organs and BMDMs were seeded at a density of $3 \times 10^5$ cells into a precoated 6 well plate with collagen. The co-cultures and the mono-cultures were grown in DMEM or 4T1-CM (as described above) and were collected after 7 days. The macrophages mono-cultured were collected at day 0 since they cannot maintain themselves in control medium, and at day 7 in cancer CM. $1 \times 10^4$ cells of fibroblasts and BMDMs were sorted using the FACSMelody instrument (BD-biosciences). All live single cells (PI negative cells after debris and doublet exclusion) were sorted. Cells staining positive for anti-CD11b-Pacific blue (Biolegend, Cat.101224) and anti-F4/80-APC Cy7 (Biolegend. cat.123117) were sorted as macrophages, and cells staining negative for these markers were sorted as fibroblasts. The cells were collected directly into lysis/binding buffer (Life Technologies), and mRNA was isolated using Dynabeads oligo (dT) (Life Technologies). Library preparation for RNA-seq (MARS-seq) was performed as previously described[49]. Libraries were sequenced on an Illumina NextSeq 500 machine and reads were aligned to the mouse reference genome (mm10) using STAR v.2.4.2a[50]. Duplicate reads were filtered if they aligned to the same base and had identical UMIs. Read count was performed with HTSeq-count[51] in union mode, and counts were normalized using DEseq2[52]. Hierarchical clustering was carried out using Pearson correlation with complete linkage, and on differentially expressed genes, which were filtered with the following parameters: basemean >5; |log fold change| > 1; FDR < 0.1. Pathway analysis was performed using Metascape[27], significant pathways were determined if $P < 0.05$, and FDR < 0.05.

### ssGSEA analysis

The ssGSEA analysis (Supplementary Data 4) was performed using GenePattern[28] on the RNAseq data. The gene signatures of protumorigenic TAMs[29] and in-vivo CAFs[30] (GEO can be accessed via #GSE195858 and #GSE195865) is based on published data and appears in Supplementary Data 4. The parameters used were gene.set.selection = ALL; sample.normalization.method = rank; weighting.exponent = 0.75; min.-gene.set.size = 10; combine.mode = combine.add.

### Ligand-receptor analysis for scRNA-seq data processing and cluster annotation

We used published scRNA-seq of breast cancer patients[14] and published scRNA-seq of 4T1 mouse breast cancer model[16]. In the human dataset, for cancer samples we focused only on the primary tumors and excluded the lymph node samples. Additionally, since the majority

of samples (32) were from female patients and only 2 samples were from male patients we excluded the male samples from our analysis. For normal tissue control, we analyzed 13 samples defined as normal. We did not analyze the preneoplastic BRCA1$^{+/-}$ samples. We filtered cells by cutoffs of gene and unique molecular identifier count greater than 200 or lower than 10,000, and a mitochondrial percentage less than 20%. We used the Seurat v.4.0.0[53] method in R v.4.2.0 for data normalization, dimensionality reduction, and clustering, using default parameters. For mouse data we subgrouped myeloid, CAFs and cancer clusters by known markers that were differentially expressed between the cultures. For human data, the Normal and male samples were removed from the analysis, we analyzed 32 patients. Shared nearest neighbor modularity optimization-based clustering was then used. Cancer, Myeloid and CAF cell clusters were selected based on classic cell markers, and selected for downstream analysis. Based on the clusters on the single cells data we used Cellchat[15] algorithm to identify the total score interactions. To further identify potential ligands between the CAF and TAMs we used NichNet[31] algorithm.

### Analysis of RARRES2 Expression in scRNA-seq
We used published scRNA-seq of breast cancer patients[14] and reanalyzed it as described above. We present the *RARRES2* gene in SCT normalization in all the clusters. To compare the expression of *RARRES2* in normal versus CAF we integrated the normal patients and subset only the fibroblast cluster based on fibroblast marker (*DCN, COL1A1*), then calculated the *RARRES2* expression per patient. In addition, based on this expression, we correlated the enrichment of the protumorigenic TAM signature[29] per patient.

### Normal epithelial cell conditioned medium
Normal epithelial cells were isolated using a similar protocol as described above for the normal mammary fat pad. After one week in culture, the cells were trypsinized and resuspended in ice-cold MACS buffer (PBS with 0.5% BSA). The samples were pelleted by centrifugation at 350 g for 5 min at 4 °C, incubated with anti-EpCAM (Miltenyi, 130-105-958) magnetic beads, transferred to LS columns (Miltenyi, 130-042-401), and enriched for epithelial cells. Then, the cells were seeded at a concentration of $300 \times 10^3$ cells/ml in 6 cm plates. After 24 h, when the cells had formed a monolayer, the medium was replaced with fresh medium. After 48 h, the medium was collected, filtered through a 0.22 μm strainer, and diluted with DMEM with 20% FBS at a ratio of 1:1.

### Macrophage count in the presence of normal epithelial and cancer CM
Mono-culture macrophages were cultured at a density of $1 \times 10^5$ cells in a pre-coated 96-well plate with collagen, in the presence of control medium (DMEM), 4T1 cancer cell CM, or normal epithelial cell CM, which were both diluted in a ratio of 1:1 with control medium. The cells were cultured for three days, and then harvested using cell titer Glo (Promega, G7572), according to the manufacturer's instructions. Luminescence was then measured using a plate reader.

### Isolation of CAFs from 4T1-GFP tumors
4T1-GFP cell line was orthotopically injected into the mammary fat pad of 8 week old BALB/c female mice ($1 \times 10^5$ cells in 50 μl PBS). After 4 weeks, the mice were sacrificed and the tumors were harvested and dissociated in gentle MACS dissociator with enzymatic solution containing 3 mg/ml collagenase (Sigma Aldrich, 11088793001) and 0.1 mg/ml Dnase in RPMI 1640 using the standard program for solid tumors. To receive single cell suspension, the digested cell suspension was filtered through 70 μ strainer with ice-cold PBS. The cells were pelleted and lysed in red blood cell lysis buffer and depleted of CD45+ and EpCAM+ cells as described above. Cells were stained for anti-CD45 FITC (Miltenyi 130-110-658), anti-CD31 FITC (Miltenyi 130-123-675), and anti-EPCAM FITC (Miltenyi 130-117-752), anti-PDPN APC (Biolegend

127410) and anti-Ly6C Pacific Blue (Biolegend 128014). See Supplementary Table 3 for antibodies information. Dead cells were excluded using PI Staining (Sigma Aldrich, P4170). CAF were gated based on PI⁻, CD45⁻, CD31⁻, EpCAM⁻, PDPN⁺. For sorting the CAF subpopulation, we used LY6C⁺ as a marker for iCAF and LY6C⁻ as a marker for myCAF. The cells were sorted using a FACSAria Fusion (BD Biosciences) into FACS tubes containing 1 ml complete DMEM media. The maximum tumor volume of 1000 (mm)$^3$ was not reached in any experiment.

### ELISA assay
The 4T1-GFP cell line and sorted CAFs were plated at a density of $7 \times 10^4$ cells in 96-well plates precoated with collagen, and cultured in RPMI 1640 supplemented with 10% FBS. After three days, the plate was centrifuged and the CM was collected. RARRES2 was detected in the CM using a mouse Chemerin/RARRES2 kit, following the manufacturer's instructions (Boster, EK1330).

### Migration assay
A total of $2.5 \times 10^5$ BMDM were seeded into 8 μm transwell inserts (Merck, PTEP24H48) in a 24-well plate and cultured for 24 h in DMEM with 2% FBS. 4T1-GFP CM was added to the bottom chamber with or without 3 nM recombinant RARRES2 (R&D, 2325-CM-025). After incubation, the inserts were removed, and the plate was centrifuged for 10 min at 4 °C. The medium was aspirated, and the cells were lysed using Glo lysis buffer (Promega, E2661) according to the manufacturer's instructions. Luminescence was measured in relative light units using a plate reader.

### Proliferation assay
A total of $1 \times 10^4$ BMDM were seeded in a 96-well plate and cultured for 24 h in DMEM with or without 3 nM recombinant RARRES2 (R&D, 2325-CM-025). The cells were lysed using cell titer Glo (Promega, G7572) according to the manufacturer's instructions. Luminescence was measured in relative light units using a plate reader.

### Real-time PCR
RNA isolation was carried using the TRIzol Reagent, based on the TRI reagent user manual (Biolab, 959758027100). Reverse transcription was done by High-Capacity cDNA reverse transcription kit (Cat 4368814, Thermo Fischer Scientific) according to the manufacturer's instructions. Quantitative RT−PCR analysis was performed using Fast SYBR Green Master mix (Applied Biosystems, 4385610) and data was normalized to the house-keeping gene HPRT. See Supplementary Table 2 for Primers list.

### Network motif analysis
The analysis began by constructing a weighted directed graph using the full interaction matrix as an adjacency matrix. Weak interactions that collectively contributed to 10% of the total interaction weights were excluded, resulting in a network with 30 edges. The root node was chosen as the node with the highest weighted outdegree. To detect 2-node motifs we used the "IGLADFindSubisomorphisms" function from the Mathematica package IgraphM to identify and enumerate all 2-cell interaction subgraphs that were isomorphic to the seven possible 2-node motifs. The average weight per edge was calculated for each subgraph instance. Summing the weight averages within each subgraph class enabled comparison between different subgraph classes. To determine the statistical significance of the identified subgraphs, we generated 10,000 degree-preserving random networks by rewiring the connections within the original network using the "IGRewire" function. Self-loops were allowed during the rewiring process. The rewiring process preserves the in-degree and out-degree of each node of the original network. The weights were randomly permuted. In each of the randomized networks, the subgraph scoring procedure was repeated for the various subgraph classes. The resulting distribution of scores

was used to calculate the z-score for each subgraph in the real versus randomized networks. Subgraphs with a z-score exceeding 1.65 were considered statistically significant. The search for 3 and 4 node motifs follows a similar procedure.

## Mathematical modeling of fibroblast-macrophage circuit

The goal of the modeling was to infer the essential factors that influence the fibroblast-macrophage circuit based on the cell count data, and to compare circuits in different contexts. These goals required a model with a minimal number of parameters to avoid overfitting. We therefore used steady-state assumptions for growth factor concentrations, leaving equations for the slower changes in cell numbers. We also incorporated detailed biochemical reactions[17,24] into a minimal number of effective interaction terms. These assumptions yielded a simple model for the rate of change of cell population, $X$, which is a balance of proliferation and removal at rates $p_X$ and $r_X$, respectively:

$$\frac{dX}{dt} = p_X X - r_X X \tag{1}$$

We describe the fibroblast-macrophage circuit by two such equations, one for each cell type ($X = F$ for fibroblasts and $X = M$ for macrophages). Fibroblasts and macrophages are removed at constant rates, $r_F$ and $r_M$, respectively. Autocrine and paracrine interactions influence the proliferation rate, $p_X$, of each cell population through exchange of growth factors ($G$). Proliferation is limited at high cell concentrations by resources in the medium and by contact inhibition. To account for this, we used a carrying capacity term ($K_X$) that makes the proliferation rate decrease with growing cell population. This logistic term originates from population ecology, and was verified for fibroblasts in vitro in ref. 17. Combining these effects yields the following equations:

$$p_X = \mu_X G_X (1 - \frac{f(F)}{K_F}) \tag{2}$$

$$\frac{dG_X}{dt} = \alpha_{XX} f(X) + \alpha_{YX} f(Y) - r_{G_X} G_X \tag{3}$$

Equation (3) describes the dynamics of the growth factor for cell population $X$. $\alpha_{XX}$ is the autocrine rate and $\alpha_{YX}$ is the paracrine rate ($Y$ denotes the other cell population). $r_{G_X}$ is the removal rate of growth factor $G_X$. The timescale of the growth factor dynamics is much faster compared to the timescale of the cell turnover. Thus, we can solve the steady state of Eq. (3) (a quasi steady state solution):

$$G_X = \frac{\alpha_{XX}}{r_{G_X}} f(X) + \frac{\alpha_{YX}}{r_{G_X}} f(Y) \tag{4}$$

and substitute it in Eq. (2) for the proliferation rate of each of the cell populations:

$$p_F = (p_{FF} f(F) + p_{MF} f(M)) \cdot \left(1 - \frac{f(F)}{K_F}\right) \tag{5}$$

$$p_M = (p_{FM} f(F) + p_{MM} f(M)) \cdot \left(1 - \frac{f(M)}{K_M}\right) \tag{6}$$

Where $p_{FF} = \frac{\mu_F \alpha_{FF}}{r_{G_F}}, p_{MF} = \frac{\mu_F \alpha_{MF}}{r_{G_F}}, p_{MM} = \frac{\mu_M \alpha_{MM}}{r_{G_M}}, p_{FM} = \frac{\mu_M \alpha_{FM}}{r_{G_M}}$ are parameter combinations that represent the autocrine and paracrine effects of the cell populations on each other.

These cellular interactions depend also on the population size, $f(X)$. Exploration of the data favored $f(X) = \log(X + 1)$, where log is the natural logarithm, which represents a nonlinear relationship with diminishing relative effects of large cell populations. This nonlinear

relationship provided better fits than a linear one, $f(X) = X$ (Supplementary Fig. 7) Adding 1 inside the log is common when working with counts, in order to avoid infinity at zero cells. The function $f$ resembles the saturation effect in Michaelis-Menten (MM) interactions. We chose not to use MM expressions to keep the lowest number of parameters possible, because each MM term requires an additional 'halfway point' parameter. Using log means that carrying capacities are in log cell numbers.

## Statistical inference

Each cell-population equation has four parameters: the rates of autocrine $p_{XX}$ and paracrine $p_{XY}$ interactions, the rate of cellular removal $r_X$, and the carrying capacity $K_X$. We sought to infer these parameters from the cell count measurements.

We divided Eq. (1) by the population size ($X$) to obtain the per-capita growth rate, which is also the logarithmic derivative: $\frac{1}{X}\frac{dX}{dt} = \frac{d\log X}{dt} = p_X - r_X$. In order to fit the data we approximated the derivative as the change in cell population over the experimental time interval ($\Delta T$): $\frac{d\log X}{dt} \simeq \frac{\Delta \log X}{\Delta T} = \frac{\log(X(t+\Delta T)) - \log X(t)}{\Delta T}$. Reordering the equation gives:

$$\log(X(t + \Delta T)) \simeq (p_X - r_X)\Delta T + \log X(t) \tag{7}$$

Taken together, each cell population number at day 7, $X_7$, can be modeled by its number and the other cell population number at day 3, $X_3$ and $Y_3$, where $\Delta T = 4$ days:

$$\log(F_7) \simeq 4\left[(p_{FF}\log(F_3+1) + p_{MF}\log(M_3+1)) \cdot \left(1 - \frac{\log(F_3+1)}{K_F}\right) - r_F\right] \\ + \log F_3 \tag{8}$$

$$\log(M_7) \simeq 4\left[(p_{FM}\log(F_3+1) + p_{MM}\log(M_3+1)) \cdot \left(1 - \frac{\log(M_3+1)}{K_M}\right) - r_M\right] \\ + \log M_3 \tag{9}$$

Each arrow from the experimental phase portrait is one sample for the fitting procedure. N arrows in a given condition provide N samples that are used to infer the 8 parameters of the corresponding model. The head of the arrow (counts in day 7) is used as the dependent variable, and the arrow's tail (counts in day 3) serves as the independent variable according to the mechanistic structure of our model.

Removal rates and carrying capacities were constrained to be positive and thus estimated by the Trust Region Reflective (TRF) method, a nonlinear least-squares approach[54]. We used the Python implementation of this algorithm, *curve fit*[55].

The experimental noise in the data led to uncertainty in parameter estimations. To estimate this, we bootstrapped (resampled the data with returns) the measurements 5000 times and inferred the parameters for each draw by the TRF algorithm. This provided a distribution for each parameter accounting for uncertainty and experimental noise. We use the parameter distributions to calculate distributions of contrasts between parameters from different conditions. Statistical significance of parameter difference was determined based on the percentile of zero (no difference) within the contrast distribution. We also used PyDREAM[26], a Bayesian tool for parameter inference, to validate our approach (Supplementary Fig. 4).

The circuit equations with inferred parameters provided streamlines on a theoretical phase portrait. We calculated the nullclines, defined as the set of points in the phase space where there is no change in the population of one cell type (dF/dt = 0 and dM/dt = 0). The intersections between these nullclines provided the fixed points of the system (Supplementary Fig. 3g, h). The net growth rate of each cell

type from the model was displayed as heatmaps (Fig. 3h–k), where the growth rate changes sign at the appropriate nullcline.

**Autocrine threshold for maintaining fibroblast population**

Under cancer CM conditions, macrophages have no inferred effect on fibroblasts growth. Therefore, the fibroblast equation is:

$$\frac{dF}{dt} = F\left( p_{FF}f(F)\cdot\left(1 - \frac{f(F)}{K_F}\right) - r_F \right) \quad (10)$$

Fibroblast population crashes when $\frac{dF}{dt} < 0$. Thus:

$$-\frac{p_{FF}}{K_F}f^2(F) + p_{FF}f(F) - r_F < 0 \quad (11)$$

To eliminate fibroblast's non-zero steady states and make their number collapse in the entire phase space, inequality (11) should hold for any $F$. This happens when $p_{FF} < \frac{4r_F}{K_F}$.

**Statistical analysis**

Statistical analysis and visualization were performed using R (Versions 3.6.0 and 4.2.0, R Foundation for Statistical Computing Vienna, Austria) and Prism 9.1.1 (Graphpad, USA). Statistical tests were performed as described in each Figure legend. In the Bulk RNA-seq, three libraries (one sample of mono-cultured fat fibroblasts and two samples of mono-cultured mammary fibroblast) were excluded due to technical problems with sequencing, as no reads were detected. In Fig. 5i, j one patient exhibited values more than 4 standard-deviations of their group mean in the RARRES2 expression, and was therefore defined as an outlier and excluded from the analysis.

**Reporting summary**

Further information on research design is available in the Nature Portfolio Reporting Summary linked to this article.

## Data availability

The breast cancer scRNA-seq datasets of human[14] and mouse[16], which were used in this study, are available on the Gene Expression Omnibus (GEO) with accession G SE161529 and at https://datadryad.org/stash/dataset/doi:10.6071/M3238R. Bulk RNA-seq data that support the findings of this study were deposited in GEO: G SE218196. The gene signatures of protumorigenic TAMs[29] and in-vivo CAFs[30] can be accessed via GEO: G SE195858 and G SE195865. The remaining data are available within the Article, Supplementary Information or Source Data file. Source data are provided with this paper.

## Code availability

We used Mathematica 13.3 for the network motif analysis. Phase portraits and parameter inference of the cell circuits were calculated using Python 3.7.4 and scipy package 1.7.3. Scripts and data needed to reconstruct the analysis and figures https://github.com/tomermilo/fibroblast_macrophage_circuits/tree/master.

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

## Acknowledgements
The authors thank all members of the RS.-S. lab and U.A. lab for their valuable input. Bioinformatic analyses were assisted by Ester Feldmesser (WIS). RS.-S. is supported by ISF grant 395/21, ERC grant 101043300, the Laura Gurwin Flug Family Fund, the Comisaroff Family Trust, the Estate of Annice Anzelewitz, the Estate of Mordecai M. Roshwal, and the David Barton Centre for Research on the Chemistry of Life. RS.-S is the incumbent of the Ernst and Kaethe Ascher Career Development Chair in Life Sciences. UA is the incumbent of the Abisch-Frenkel Professorial Chair and is supported by a Cancer Research UK grant (C19767/A27145). ET is supported by grants from the Israel Science Foundation and the EU Horizon 2020 Research and Innovation Program REANIMA.

## Author contributions
S.Ma. and T.M. designed, performed and analyzed experiments and wrote the manuscript. A.I., C.H., C.L. and M.P.F designed and performed experiments. T.M and A.M designed and performed mathematical modeling and statistical analysis. S.Mi., and E.T. provided intellectual input. C.L. and Y.S. directed and designed bioinformatic analysis. U.A and R.S.-S. designed and directed the study, designed and analyzed experiments, secured funding and wrote the manuscript.

## Competing interests
The authors declare no competing interests.
