## [Peer Review File · Nature Communications]

The tumor microenvironment shows a hierarchy of cell-cell interactions dominated by fibroblastsReviewers' comments:

Reviewer #1 (Remarks to the Author): with expertise in mathematical modelling for cellular biology

This is an interesting paper by Mayer et al., that aims to employ a "circuit" based abstraction to model the interactions between fibroblasts and macrophages in a (tumor) microenvironment. The authors first carry out some experiments to explore macrophage (M) and fibroblast (F) interactions and how different abundances of one affect the other in terms of growth and survival. The authors then employ media that was used to grow tumor cells and use this media to explore how M and F interact in the context of tumor-associated media. The authors first build a "phase diagram" of cell growth, then derive some modeling interactions based on the data, and then use RNASeq gene expression profiling to identify the differences and similarities about this work.

Overall I found the general idea of the paper reasonably interesting but lacking in innovation. The main conclusions about the differences between cell growth in the presence or absence of cancer-cell derived media is interesting, but merits a significant amount of extra work. I discuss my perceived major and minor flaws in this paper below:

MAJOR FLAWS:

1) I think that the idea of exploring the interactions between Macrophages and Fibroblasts is a good one, particularly given that these interactions can model various real-world interactions such as tumor microenvironment (TME). Having only two time points (day 3 and 7) does not seem nearly enough to capture the full dynamic range of a complex set of interactions. For example, prior work has carried this out in 2D and 3D environments with various forms of microscopy and analytics and found that the dynamics of the system can be quite rich. Although a first order approximation seems reasonable, I worry that the authors may be missing non-linear effects that could emerge from complex cell-cell interactions.

2) The model construction should be better supported by the biology. As shown in Figure 2 A and B, the model comprised a growth rate, a death rate, some modulating carrying

capacity (was this needed?), and a set of $F \leftrightarrow M$ interactions that were not clearly described. I understand that a central goal of the model was to keep things simple, but this seems like an overly simplified formulation of the problem. I would imagine, at the very least, that e.g. Hill function formulation of some of these processes to account for many possible underlying interactions would e.g. remove the dependence of the model on "carrying capacity" rates that seem to have been introduced in an ad hoc manner. IN the end, I am not sure we learn that much from an overly simplified model that generates the results that were coded into it.

3) The model calibration, for a set of models with various model parameters, lacks robustness. The authors did not describe their model parameter calibration or inference in much detail, other than to state that they used a curve fitting approach using a Trust Region Reflective method. What was the objective or scoring function used by the authors? What was minimized? My understanding is that the models in Equations 5 and 6 would each give rise to one of the arrows in the Figure 2. Does this mean that multiple parameters yielded a distribution? Given the complexity of the data and multiple parameter sets found in the model, why not use a technique such as Bayesian inference that actually yields parameter distributions, and from which one can learn things like sensitivity to a given parameter and whether the parameter was indeed identifiable? I would feel much more comfortable believing the model if I knew which parameters were identifiable or not, and whether their posterior distributions diverge? Several tools have come up in recent years for parameter estimation, such as PyDREAM (Shockley et al., bioinformatics) or PyBioNetFit (Mitra et al, iScience).

4) The authors then employ Akaike information criterion (AIC) to identify the models that best describe the data. Since the authors are effectively carrying out model selection (which they do not discuss much in the intro), I would imagine that they are determining the dominant parameters via sums of weights? Or simply by Akaike score? An interesting body of work (for example Galipaud et al., papers in *Methods in Ecology and Evolution* but many others) have discussed the flaws associated with AIC for model selection and state that it should at the very least should be used with great caution. I am simply not convinced from the data in Figure S3 that the AIC was estimated in a convincing manner. How were the AIC

weights corrected? Again, a Bayesian approach would yield much more insightful results rather than ones limited to a single "best fit". There certainly could be overlap between the parameters...

MINOR Weaknesses:

Various typos throughout. For example, in the abstract, do the authors mean to say "comprised" instead of composed?

some of the figures are difficult to interpret. In figure 3 the labels are very small. Would Figure 3 G,H,I,J better be presented as a table?

Reviewer #2 (Remarks to the Author): with expertise in computational modelling

In this manuscript, the authors utilized a co-culture of macrophages and organ-derived fibroblasts to dissect organ specific cell circuits and explore circuit behaviors in cancer-conditioned medium. The highlights of this study are (1) to fit mathematical models of cell circuits against temporal cell population dynamics derived from flow cytometry, (2) to compare circuits models from various conditions and illustrate the supporting evidence for the modeling outcomes using RNA-seq data, (3) to explore more details of cell-cell interactions by exploring biological pathways and ligand-receptor interactions using bulk and single cell RNA-seq data. Overall, the work was nicely done to integrate systems biology modeling and bioinformatics. To improve the quality of the article, the authors need to address the following questions.

(1) The authors measured cell counts for 3d and 7d, and later for 14d and 21d. I wonder why the authors didn't use the cell counts for the initial conditions (0d). I asked this because even for 3d the cell count data were clustered in narrowed regions in the phase plane (counts of fibroblasts vs. macrophages). My guess is that the initial conditions were widespread, but because of the temporal dynamics during 0 – 3 days, the tested conditions at 3d are much narrowed. Thus, cell counts for many regions were not accessible anymore

at 3d.

(2) In Figure 1B-G, how were the steady states defined by the vectors? It seems that the states were derived from Figure 2C-F, although it was not mentioned in Figure 1, and not all the states from Figure 2 were listed in Figure 1. The authors need to clarify if the states were obtained from model fitting, or they were defined directly by the vectors. Because of the sparsity of the “vector field”, to me it is hard to determine the separatrix and in turn the unstable/saddle steady states. In fact, some vectors in lung/fat/cancer conditions were clearly opposite to the directions that were supposed to be defined by the assigned steady states. The authors have already provided the goodness of fitting by plotting observed and predicted cell counts in day 7 (in the logscale though) in Figure S3 I-P. It would be important to also evaluate the direction of the changes from observation and prediction.

(3) In page 8, the authors mentioned “standard information criteria”. It is beneficial to refer to the AIC methods from the Method section.

(4) In the results of Figure 3O-R and Figure S3 A-H, the authors evaluated AIC of the full model and the models with one less parameter. I wonder why the elimination of parameter is only limited by one. Also, it’s hard to determine the best model for cases when the multiple AIC scores are close to the minimum (e.g., Figure S3A, Full and -P_FM). Would the parameter fitting work well when P_FM is also eliminated for mammary? In addition, I believe the labels P_FM and P_MF were swapped in Figure S3A-H (I used in the original labels in the previous sentences).

(5) In Figure 3AEF, the authors show nicely the pathways for different gene clusters from macrophages. I suggest the authors also do so for gene clusters from fibroblast (Figure S5B).

(6) In Figure 4AB, ICELLNET ligand-receptor analysis was applied to RNA-seq data for the mammary circuits. Could this be performed for the other two organs? I wonder if the M to F interactions were significantly weaker in these cases.

(7) In page 22 the third paragraph, it seems that Figure S6G was wrongly cited.

(8) In the modeling equations (2-3) in page 28, if $f(X)$ is chosen to be the logarithm function (btw, is that \log_{10} ?), then K_F and K_M would have different meanings than carrying capacity. The terms have been used in log scale and non-logscale in multiple places (e.g. throughout Figure 2).

(9) Related to the previous question: the authors mentioned “This nonlinear relationship provides better fits than a linear one”. Could the authors provide quantitative evidence?

(10) In data availability, the bulk RNAseq data was described as deposited, but it is still not accessible. Also, the authors need to deposit the cell cytometry data, at least the cell counts table for various conditions.

Reviewer #3 (Remarks to the Author): with expertise in mathematical biology

The authors present an integrative work regarding the identification of macrophages and fibroblasts interactions in the tumor microenvironment. The work involves in vitro experiments with murine cells where co-cultures of fibroblasts and macrophages were embedded in normal and cancer microenvironments. In turn, a mathematical model was fitted to experiments. Finally, transcriptional profiling of CAFs taken from mice and humans show similar traits to the ones identified in the in vitro experiments. The work is very interesting but before publication I would like the authors to address the following points:

- The in vitro model reflects the situation where resident macrophages interact with fibroblasts. However, it is known that in vivo most of macrophages are recruited from the blood stream. It would be interesting to know how this would impact the system. Certainly, this could be addressed in the mathematical model.
- Regarding the parameter calibration I was wondering if this parameter value selection is a local minimum. Does the algorithm performs any perturbation to identify the robustness of the parameter selections? Another interesting point would be to analyze the sloppiness (literature from J. Senth) of the system and find out which parameters impact the system behavior the most. Alternatively, one could make bifurcation diagrams from all parameters, since they are not so many.
- I know that the authors didn't study the impact of other determinants such as hypoxia or inflammation (and the consequent polarization of the macrophages) but they could discuss it maybe in the light of a recent paper Setten et al, Nat. Comms, 2022.
- Finally a paragraph with the study limitations would be very useful.

Reviewer #4 (Remarks to the Author): with expertise in fibroblasts/macrophages cross-talk, cell-to-cell communication

Cell circuits and fibroblasts and macrophages in the tumour microenvironment.

Here the authors measure population dynamics of directly co-cultured macrophages and fibroblasts to identify regulators of cell circuits. The authors observe that whereas monocultured fibroblasts are self-sustained, macrophages are not and rely on interactions with co-cultured fibroblasts. Moreover, and curiously, the authors identify minor differences in the interactions between macrophages and fibroblasts from breast, mesometrium and the lung. More significantly, tumour cell conditioned medium offset the population dynamics. Finally, the authors use transcriptional and computational analysis to identify putative signals regulating the cell circuits.

Overall, the manuscript is nicely presented and the work is of high quality. However, the conceptual advances are minor and there are limited direct molecular evidence included to substantiate the authors' claims.

Specifically:

The authors describe how fibroblasts and macrophages differ in their interdependencies. However, these experiments are done in the presence of 10%FBS, which acts as a significant supporter of fibroblasts but perhaps not as much for macrophages. The point being that the experimental setup may be biased in support of their observations and consequently raises concerns about the meaningfulness of these.

Similarly, the authors use conditioned medium from a cancer cell line and claim this disrupts the observed cell circuitries. However no normal control is used for comparison. Moreover, tumour cells exist within a cellular complex environment with a resulting change to the tumour cell transcriptional phenotype e.g. tumour cell conditioned medium does not take into account the effects of host cells on tumour cell gene expression. Finally, there is also a distinct possibility the observed interactions are due to differences in metabolic requirements, but this is not even discussed.

The authors present some very interesting quantitative projections of the cell circuitries to identify differences between interactions between macrophages and fibroblasts from different origins. However, it is difficult to estimate whether small difference between numbers e.g. 0 and 0.02 (Fig 2o-r) are functionally meaningful without quantitative validation.

The authors use RNA expression analysis to identify putative ligand-receptor pairs that may mediate the differences in cell interactions. Firstly, there is no functional analysis to confirm that indeed differences in paracrine cell signalling is a major contributor compared to other changes such as metabolic capacity. Moreover, the functional relevance of the predicted interactions are not validated and thus it remains unclear whether the observed expression changes are related to the described cell circuitries. Moreover, the authors note that macrophage gene expression is more strongly affected by fibroblast co-culture than fibroblasts by macrophage co-culture. This could also be a reflection of distinct sensitivities to the growth conditions eg DMEM/FBS impacts fibroblast gene expression to a larger extent than it affects macrophage gene expression and thus fibroblasts appear less sensitive to the co-cultures than macrophages.

Point-by-point response to the reviewer's comments:

Reviewer #1 (Remarks to the Author): with expertise in mathematical modeling for cellular biology

This is an interesting paper by Mayer et al., that aims to employ a "circuit" based abstraction to model the interactions between fibroblasts and macrophages in a (tumor) microenvironment. The authors first carry out some experiments to explore macrophage (M) and fibroblast (F) interactions and how different abundances of one affect the other in terms of growth and survival. The authors then employ media that was used to grow tumor cells and use this media to explore how M and F interact in the context of tumor-associated media. The authors first build a "phase diagram" of cell growth, then derive some modeling interactions based on the data, and then use RNASeq gene expression profiling to identify the differences and similarities about this work.

Overall I found the general idea of the paper reasonably interesting but lacking in innovation. The main conclusions about the differences between cell growth in the presence or absence of cancer-cell derived media is interesting, but merits a significant amount of extra work. I discuss my perceived major and minor flaws in this paper below:

Response: We thank the reviewer for providing helpful comments. We have made a significant addition to our manuscript that enhances the novelty of our study. Specifically, we included a circuit motif analysis of the network of all interactions between different cell types in the tumor microenvironment. Surprisingly, we found that one circuit motif was far more prevalent than others and appeared repeatedly with different cell types. This circuit motif includes an autocrine loop of two cell types and bi-directional paracrine interactions with another cell type, which precisely matches the motif observed in our in-vitro model. Of all possible options, the most abundant instance of this motif is the CAF-TAM interaction. The new network motif analysis therefore provides a framework and rationale for focusing on CAF-TAM interactions (Figure 1, see below with the related text, lines 77-120).

Our revised manuscript also more clearly emphasizes the conceptual advance over earlier work. We offer a new concept for simplifying the tumor microenvironment into small circuits of a few cell types which can be studied *in-vitro*, while retaining their *in-vivo* relevance. This allows characterizing novel interaction molecules, and understanding the dynamics and bistability properties of the TME at unprecedented quantitative resolution.

We further use the network analysis and circuit approach to identify a novel CAF-TAM interaction between RARRES2 and its receptor CMKLR1. We show that RARRES2 enhances macrophage migration and has clinical relevance in human data (Figure 5, see below with the related text, lines 419-464).

Overall, these findings extend far beyond what we originally anticipated and provide novel insights into the biology of the tumor microenvironment.

Figure 1: Network analysis reveals a hierarchy of interactions with a dominant CAF-TAM circuit in the breast tumor microenvironment. **A.** UMAP visualization of the main cell clusters in the breast TME in human scRNA-seq data from 32 patients ¹⁴. **B.** Heatmap of interaction strengths between pairs of cells based on cumulative ligand-receptor interaction scores using CellChat ¹⁵ applied to the scRNA-seq data of **(A)**. **C.** Illustration of the structure of the network based on the analysis in **(B)** shows hierarchy (see Methods). Arrow width is proportional to the interaction score. **D.** Network motif analysis of all 7 possible two-cell circuit patterns, tested for abundance relative to randomized degree-preserving networks. The dashed line represents a 0.05 p-value threshold. **E.** The top three two-cell subgraphs by the average weight all include CAFs. **F.** Network motif analysis of the 13 possible three-cell-circuit patterns. The dashed line represents a 0.05 p-value threshold. **G.** The top three scoring three-cell circuit subgraphs. **H.** The three-cell circuit of CAFs, TAMs and cancer cells.

“Analysis of the breast cancer microenvironment reveals a hierarchy of interactions with a dominant CAF-TAM circuit

To begin to untangle the complexity of the TME, we mapped the network of cell-cell interactions by analyzing published scRNA-seq data from breast cancer patients ¹⁴. We identified the cell types (Figure 1A and S1A) and scored their interactions (Supplementary Table 1) using CellChat ¹⁵, a tool for estimating ligand-receptor interaction strength. We found multiple pairs of interacting cell types (Figure 1B), and used CellChat to score the strength of each interaction.

We found that the strongest interaction occurred between CAFs and myeloid cells (comprised mainly of tumor-associated macrophages, hereafter referred to as TAMs), followed by the interaction between CAFs and mast cells (Figure 1B), and an autocrine loop in which CAFs send ligands which they also sense by expressing their receptors. In fact, CAFs were the cell type with the highest interaction scores. These features are found also in mouse breast cancer scRNA-seq data ¹⁶. In mice, as in human data, CAFs are the most interacting cell type and their autocrine interaction, as well as their interaction with macrophages, are among the strongest (Figure 1B, S1B).

To better understand the structure of the interactions among the cell types, we constructed a weighted and directed cell-network from the interaction matrix (Figure 1C, S1C; See Methods for details). This analysis revealed that the interaction network is hierarchical and mostly feedforward, with CAFs at the top sending signals to the other cell types, and TAMs at the bottom, receiving signals from CAFs and other cell types (Figure 1C).

We next asked whether the complex network of interactions can be simplified and described in terms of repeating instances of smaller circuits, which we can isolate and further explore. For this purpose we employed network motif analysis, which detects small circuits that occur in the network significantly more often than in randomized networks. We began with circuits made of two interacting cell types. Of the 7 possible connected two-cell circuits (Figure 1D), we found that only one type recurs with an interaction score that exceeds those found in randomized networks, and is thus a network motif ^{12,13}. This circuit has two interacting cell types that send mutual paracrine signals, and each also has an autocrine signaling loop. This circuit appears three times in the network, and all have CAFs as one of the nodes. Of these circuits the strongest circuit in terms of total and mean interaction score is the CAF-TAM circuit (Figure 1E).

We also analyzed circuits made of three and four cell types (Figure 1F-G, S1D). The most common circuits were made of combinations of the above-mentioned two-cell circuit (Figure 1E). The CAF-TAM pair participated in the highest scoring instances of these three- and four-cell circuits (Figure 1G, S1D). For example, the CAF-TAM circuit interacts with cancer cells to form a three-cell circuit in which CAFs send signals to cancer cells which in turn send signals to TAMs (Figure 1H).

We conclude that the breast cancer TME interaction network is hierarchical and composed of repeated occurrences of a specific two-cell circuit motif. CAFs are at the top of the hierarchy and send out many signals, whereas TAMs are at the bottom receiving end. The CAF-TAM two-cell circuit is an example of this motif and is one of the circuits with the strongest interactions in the network. Thus, studying this circuit in isolation can improve our understanding of the TME interaction network.”

Figure 5: RARRES2 and its receptor CMKLR1 are part of the CAF-TAM signaling axis in breast cancer. A. Venn diagram of potential CAF-to-TAM ligands shared between breast cancer patient tumors (Pal et al., 2021) and 4T1 tumors in mice (Sebastian et al., 2020a) based on the Nichnet ligand-receptor tool. **B.** Heatmap representation of the expression of the 77 shared ligands from (A) in fibroblasts from the *in vitro* cell-circuit, based on bulk RNA-seq data from Figure 4B. **C-D.** qPCR of *Rarres2* and *Cmlklr1* genes from fibroblasts and macrophages monocultured or co-cultured in control or cancer CM for 72 hours. **E.** CAFs and cancer cells from 4T1 tumors grown in syngeneic mice were isolated, cultured for 72h and RARRES2 levels in the media were assessed by ELISA. **F.** Transwell migration

assay of macrophages in the presence of control medium, or 4T1 cancer CM with or without recombinant RARRES2 for 24h. Luminescence values were normalized to 4T1-CM in log2. Data are combined from at least three independent experiments, with a total of n=9-12 biological replicates. P-value was calculated using one-way ANOVA followed by Tukey's multiple comparisons test. Error bars represent SEM, *p < 0.05, **p < 0.01, ***p < 0.001, ****p < 0.0001. **G.** Violin plot of RARRES2 expression based on human scRNA-seq of normal fibroblasts (N=17) versus CAFs (N=32). P-value was calculated using Students' T-test, ****p < 0.0001. **H.** RARRES2 expression in different clusters from human scRNA-seq data, including CAF subclusters (defined using markers shown in Figure S5A). **I.** Pearson correlation between fibroblasts expressing high levels of RARRES2 and macrophages expressing high levels of the pro-tumorigenic TAM signature. Each dot represents one patient, based on human scRNA-seq (Pal et al., 2021). **J.** RARRES2 gene expression in breast cancer patients (Pal et al., 2021) stratified by grade (high grade = grade 3, N=21; low-grade = grade 1 and 2, N=10). P-value was calculated using Students' T-test, **p < 0.01.

“The isolated circuit highlights RARRES2 and CMKLR1 as potential mediators of CAF-TAM signaling

The isolated cell circuit enables testing the effect of specific ligand-receptor interactions on the dynamics, function and transcriptomes of the circuit. We therefore asked which ligands are upregulated in both human and mouse CAFs, as well as in the co-culture. First, we identified shared ligands between human and mouse breast CAFs based on scRNA-seq datasets ^{14,16}, using the NicheNet tool ³². This analysis revealed 77 shared ligands which have potential receptors on TAMs (Figure 5A). Next, we assessed the expression of these ligands in the RNA-seq data from our co-culture (Figure 5B). We identified a cluster of 23 ligands upregulated in the presence of cancer CM (Figure 5B, cluster 1). We validated the expression of several genes from this cluster by qRT-PCR and found that indeed they were all significantly upregulated in fibroblasts co-cultured in the presence of cancer CM (Figure 5C, S5C).

This cluster included genes well known to mediate CAF-TAM interactions in cancer, such as Tgfb1, Csf1, and Ccl2 ³³. It also included Retinoic Acid Receptor Responder (Rarres2; also known as Chemerin), a chemokine known to regulate inflammation, adipogenesis, and metabolism through activation of the chemokine-like receptor 1 (CMKLR1) ^{34,35}. RARRES2 was recently found to be expressed by CAFs in colorectal cancer ³⁶. However, its role in the TME was not elucidated. We therefore decided to focus our analyses on RARRES2. We found that Rarres2 is upregulated in fibroblasts cultured in the presence of cancer CM, and its expression is further induced by co-culture with macrophages (Figure 5C). Strikingly, its receptor, Cmk1r1, is upregulated in macrophages upon co-culture with fibroblasts or in the presence of cancer CM (Figure 5D). To directly test whether RARRES2 is not only expressed by CAFs but also secreted by them, we isolated CAFs from 4T1 tumors, cultured them for 3 days and measured the levels of RARRES2 in the medium by ELISA. We also isolated and cultured the 4T1 cancer cells themselves, as control (Figure 5E). This analysis confirmed that RARRES2 is secreted from CAFs and not from cancer cells. To gain insight into how RARRES2 affects TAMs, we monitored macrophage proliferation and migration in the presence of control medium, cancer CM, and cancer CM with recombinant RARRES2. RARRES2 did not affect macrophage proliferation (Figure S5D). It did however significantly enhance macrophage migration (Figure 5F), highlighting the potential role of RARRES2-CMKLR1 signaling in breast cancer.

To test the relevance of our findings to human disease, we compared RARRES2 expression in CAFs to fibroblasts from normal breast. Analysis of the human breast scRNA-seq dataset ¹⁴ showed that RARRES2 is specifically upregulated by CAFs and not by normal mammary fibroblasts (Figure 5G). Moreover, it is uniquely expressed by CAFs and not by other cell types within the human breast TME (Figure 5H, S5A). Next, to explore the potential effects of RARRES2 expression on TAMs in patients, we computed the correlation between CAFs expressing high

levels of RARRES2 and macrophages expressing a protumorigenic TAM signature associated with poor survival ²⁹ in the human breast cancer RNA-seq dataset ¹⁴. We found a strong correlation, consistent with RARRES2 expression contributing to a protumoral TAM phenotype (Figure 5I). To further assess the clinical relevance of RARRES2 expression, we compared the expression levels of RARRES2 in low-grade vs high-grade tumors. We found that RARRES2 expression is significantly higher in high-grade cases compared to low-grade cases (Figure 5J). We conclude that RARRES2, identified through our in-vitro circuit approach, mediates CAF signaling to TAMs in mouse models of breast cancer and in human disease, and may serve as a therapeutic target for future exploration.”

MAJOR FLAWS:

1) I think that the idea of exploring the interactions between Macrophages and Fibroblasts is a good one, particularly given that these interactions can model various real-world interactions such as tumor microenvironment (TME). Having only two time points (day 3 and 7) does not seem nearly enough to capture the full dynamic range of a complex set of interactions. For example, prior work has carried this out in 2D and 3D environments with various forms of microscopy and analytics and found that the dynamics of the system can be quite rich. Although a first order approximation seems reasonable, I worry that the authors may be missing non-linear effects that could emerge from complex cell-cell interactions.

Response: Thank you for this comment, which helped us clarify the text. The crucial point which we now emphasize is that one can infer the system dynamics by measuring a sufficient amount of different initial conditions along the phase plane. This is equivalent to measuring multiple time points along a single trajectory.

We now added the following text to the results section:

Lines 271-273: “We estimated the values of the model parameters by fitting cell numbers at day 7 given their numbers at day 3 (Figure 3C-E, see Methods). Many distributed different initial conditions allowed us to infer the dynamics for any initial condition at all timepoints.”

and to the methods section:

Lines 910-914: “Each arrow from the experimental phase portrait is one sample for the fitting procedure. N arrows in a given condition provide N samples that are used to infer the 8 parameters of the corresponding model. The head of the arrow (counts in day 7) is used as the dependent variable, and the arrow’s tail (counts in day 3) serves as the independent variable according to the mechanistic structure of our model.”

In addition, we selected a small number of initial conditions and tested two additional timepoints beyond day 3 and 7 (day 14, 21, Figure S2E). We find that the appropriate arrows align in the phase portrait, which indicates that there is time invariance in the dynamics to a good approximation.

Supplementary Figure 2E. An experimental phase portrait comparing dynamics at different time points - co-cultures assayed at days 7 to 14 are represented by orange arrows; co-cultures assayed at days 14 to 21 are represented by green arrows. These are overlaid on the experimental phase portrait presented in Figure 1C (gray arrows).

2) The model construction should be better supported by the biology. As shown in Figure 2 A and B, the model comprised a growth rate, a death rate, some modulating carrying capacity (was this needed?), and a set of $F \leftrightarrow M$ interactions that were not clearly described. I understand that a central goal of the model was to keep things simple, but this seems like an overly simplified formulation of the problem. I would imagine, at the very least, that e.g. Hill function formulation of some of these processes to account for many possible underlying interactions would e.g. remove the dependence of the model on "carrying capacity" rates that seem to have been introduced in an ad hoc manner. IN the end, I am not sure we learn that much from an overly simplified model that generates the results that were coded into it.

Response: We thank the reviewer for this comment, which prompted us to better describe how our model construction is supported by the biology.

- We clarify the role of the carrying capacity. Without the carrying capacity the steady state cannot be stable because the circuit is essentially a positive feedback loop between cells that provide each other with growth factors. Carrying capacity is a nonlinear term that prevents cell populations from growing to infinity. It is experimentally motivated - carrying capacity for fibroblasts was experimentally demonstrated in Zhou et al. Cell, 2018.

This clarification was added to the result section:

Lines 266-268: "Fibroblast numbers cannot exceed a carrying capacity K_F - the maximal cell population that prohibits further growth - which is determined by environmental factors such as nutrients and space availability^{17,25}."

and to the Methods section:

Lines 864-867: "Proliferation is limited at high cell concentrations by resources in the medium and by contact inhibition. To account for this, we used a carrying capacity term (K_X) that makes the proliferation rate decrease with growing cell population. This logistic term originates from population ecology, and was verified for fibroblasts in vitro by Zhou et al¹⁷."

- We clarify in Methods that the mathematical forms of the paracrine interactions (F<>M) were derived from a quasi-steady-state solution (using separation of timescales) of the equations of the growth factor (G) dynamics: $\frac{dG_X}{dt} = \alpha_{XX}f(X) + \alpha_{YX}f(Y) - r_{G_X}G_X \Rightarrow G_X = \frac{\alpha_{XX}}{r_{G_X}}f(X) + \frac{\alpha_{YX}}{r_{G_X}}f(Y)$. We substitute this quasi state solution in the equations of the cell population dynamics.

This clarification was added to the text:

Lines 874-878: “The timescale of the growth factor dynamics is much faster compared to the timescale of the cell turnover. Thus, we can solve the steady state of equation (3) (a quasi steady state solution):

$$(4) G_X = \frac{\alpha_{XX}}{r_{G_X}}f(X) + \frac{\alpha_{YX}}{r_{G_X}}f(Y)$$

and substitute it in equation (2) for the proliferation rate of each of the cell populations”

- Our goal in developing the current model was to allow parameter identifiability. This is essential to be able to infer their magnitude and compare different conditions. The circuit models in previous publications (Zhou et al. 2018) were not fully identifiable. Our new model therefore provides a significant conceptual advance and opens the path to new biological discoveries.
- For simplicity and identifiability we minimized the number of model parameters. This is why we decided to approximate Michaelis–Menten kinetics with logarithmic functions.
Lines 887-890: “The function f resembles the saturation effect in Michaelis-Menten (MM) interactions. We chose not to use MM expressions to keep the lowest number of parameters possible, because each MM term requires an additional ‘halfway point’ parameter.”
- The main conclusion on the inferred interaction strengths was not built into the model. Rather, the hierarchy of interaction strengths was a novel prediction that arose from the model, and which we tested and validated using RNA-seq. It is biologically important because it provides potential targets for intervention.

3) The model calibration, for a set of models with various model parameters, lacks robustness. The authors did not describe their model parameter calibration or inference in much detail, other than to state that they used a curve fitting approach using a Trust Region Reflective method. What was the objective or scoring function used by the authors? What was minimized? My understanding is that the models in Equations 5 and 6 would each give rise to one of the arrows in the Figure 2. Does this mean that multiple parameters yielded a distribution? Given the complexity of the data and multiple parameter sets found in the model, why not use a technique such as Bayesian inference that actually yields parameter distributions, and from which one can learn things like sensitivity to a given parameter and whether the parameter was indeed identifiable? I would feel much more comfortable believing the model if I knew which parameters were identifiable or not, and whether their posterior distributions diverge? Several tools have

come up in recent years for parameter estimation, such as PyDREAM (Shockley et al., bioinformatics) or PyBioNetFit (Mitra et al, iScience).

Response: Following the reviewer’s suggestion we validated our model parameter inference with the Bayesian tool, PyDREAM. Five sampling Markov chains converged and were well-mixed (Appendix 1 to this letter). We obtained similar parameter distributions (Appendix 2) and thus conclude that our inference is robust.

Appendix 1: Validation of parameter calibration with Bayesian inference implemented in the python package PyDREAM (Shockley et al., 2018). Trace plots of five independent Markov chains for each condition (control at the top two rows and cancer at the bottom two) with zoom (insets) on the last 2,000 samples to show convergence.

Appendix 2: Parameter distributions, pooled from all 2,000 last samples of the five Markov chains. The parameter distributions are similar to the ones in Figure 3C obtained by bootstrapping.

The comment also helped us to clarify our original approach, as follows:

The objective function was least squares and the Trust Region Reflective algorithm was used to estimate its minimum under bounded parameters constrictions - we demanded non-negative carrying capacities and removal rates. We used a well-established software python package, SciPy (Virtanen et al., Nature Methods, 2020).

This clarification was added to the text:

Lines 910-923: “Each arrow from the experimental phase portrait is one sample for the fitting procedure. N arrows in a given condition provide N samples that are used to infer the 8 parameters of the corresponding model. The head of the arrow (counts in day 7) is used as the dependent variable, and the arrow’s tail (counts in day 3) serves as the independent variable according to the mechanistic structure of our model.

Removal rates and carrying capacities were constrained to be positive and thus estimated by the Trust Region Reflective (TRF) method, a nonlinear least-squares approach⁵⁶. We used the Python implementation of this algorithm, `curve_fit`⁵⁷.

The experimental noise of the data led to uncertainty in parameter estimations. To estimate this, we bootstrapped (resampling the data with returns) the measurements 5,000 times and inferred the parameters for each draw by the TRF algorithm. This provided a distribution for each parameter accounting for uncertainty and experimental noise.”

4) The authors then employ Akaike information criterion (AIC) to identify the models that best describe the data. Since the authors are effectively carrying out model selection (which they do not discuss much in the intro), I would imagine that they are determining the dominant parameters via sums of weights? Or simply by Akaike score? An interesting body of work (for

example Galipaud et al., papers in Methods in Ecology and Evo but many others) have discussed the flaws associated with AIC for model selection and state that it should at the very least should be used with great caution. I am simply not convinced from the data in Figure S3 that the AIC was estimated in a convincing manner. How were the AIC weights corrected? Again, a Bayesian approach would yield much more insightful results rather than ones limited to a single "best fit". There certainly could be overlap between the parameters...

Response: We thank the reviewer - this comment helped us to remove model selection from the paper altogether - a weak part in our analysis which was tangential to the main point of the paper. We now use a single model (instead of many models that differ in terms of setting one parameter to zero), and estimate all parameter distributions. This removes the need for model selection criteria. All conclusions remain essentially the same, especially the hierarchy of interaction strengths where the autocrine loop of fibroblasts is strongest. The rigor of the analysis is much improved. Moreover, as mentioned in our response to comment (3), conclusions made by Bayesian tools converge to ours.

MINOR Weaknesses:

Various typos throughout. For example, in the abstract, do the authors mean to say "comprised" instead of composed?

Response: Thank you, we fixed all typos.

some of the figures are difficult to interpret. In figure 3 the labels are very small. Would Figure 3 G,H,I,J better be presented as a table?

Response: These issues were fixed.

Reviewer #2 (Remarks to the Author): with expertise in computational modeling

In this manuscript, the authors utilized a co-culture of macrophages and organ-derived fibroblasts to dissect organ specific cell circuits and explore circuit behaviors in cancer-conditioned medium. The highlights of this study are (1) to fit mathematical models of cell circuits against temporal cell population dynamics derived from flow cytometry, (2) to compare circuits models from various conditions and illustrate the supporting evidence for the modeling outcomes using RNA-seq data, (3) to explore more details of cell-cell interactions by exploring biological pathways and ligand-receptor interactions using bulk and single cell RNA-seq data. Overall, the work was nicely done to integrate systems biology modeling and bioinformatics.

Response: We thank the reviewer for this endorsement. Supported by the comments and suggestions we performed additional computational analysis and functional experiments that led us to focus, in our revised manuscript, on the effects of cancer on the fibroblast-macrophage

circuit. Much of the organ-related data was moved to Supplementary sections, and a new network analysis of the breast tumor microenvironment was added.

Specifically, we define the full network of interactions between cell types in the breast cancer TME in an unbiased manner using human scRNA-seq data and ligand-receptor interaction scores. We find that the network is hierarchical with CAFs at the top of the hierarchy. We used network-motif analysis to identify a recurring two-cell circuit motif and found that the strongest motif in the network is precisely the circuit that we study in the paper - mutual paracrine interactions between two cell types, each of which also has an autocrine loop. The strongest instance of this motif is the CAF-TAM interaction. The new network motif analysis therefore provides a framework and rationale for focusing on CAF-TAM interactions (Figure 1, see below with the related text, lines 77-120).

We further use the network analysis and circuit approach to identify a novel CAF-TAM interaction between RARRES2 and its receptor CMKLR1. We show that RARRES2 enhances macrophage migration, and has clinical relevance in human data (Figure 5, see below with the related text, lines 419-464).

Finally, we apply a revised mathematical approach in which we bypass model selection altogether by using a single biologically-motivated model. We also omitted much of the cross-organ comparisons, and focus instead on cancer vs normal breast circuits, where the effect size is greater.

Figure 1: Network analysis reveals a hierarchy of interactions with a dominant CAF-TAM circuit in the breast tumor microenvironment. **A.** UMAP visualization of the main cell clusters in the breast TME in human scRNA-seq data from 32 patients¹⁴. **B.** Heatmap of interaction strengths between pairs of cells based on cumulative ligand-receptor interaction scores using CellChat¹⁵ applied to the scRNA-seq data of (A). **C.** Illustration of the structure of the network based on the analysis in (B) shows hierarchy (see Methods). Arrow width is proportional to the interaction score. **D.** Network motif analysis of all 7 possible two-cell circuit patterns, tested for abundance relative to randomized degree-preserving networks. The dashed line represents a 0.05 p-value threshold. **E.** The top three two-cell interaction subgraphs scored by the average weight all include CAFs. **F.** Network motif analysis of the 13 possible three-cell-circuit patterns. The dashed line represents a 0.05 p-value threshold. **G.** The top three scoring three-cell circuit subgraphs. **H.** The three-cell circuit of CAFs, TAMs and cancer cells.

“Analysis of the breast cancer microenvironment reveals a hierarchy of interactions with a dominant CAF-TAM circuit

To begin to untangle the complexity of the TME, we mapped the network of cell-cell interactions by analyzing published scRNA-seq data from breast cancer patients ¹⁴. We identified the cell types (Figure 1A and S1A) and scored their interactions (Supplementary Table 1) using CellChat ¹⁵, a tool for estimating ligand-receptor interaction strength. We found multiple pairs of interacting cell types (Figure 1B), and used CellChat to score the strength of each interaction.

We found that the strongest interaction occurred between CAFs and myeloid cells (comprised mainly of tumor-associated macrophages, hereafter referred to as TAMs), followed by the interaction between CAFs and mast cells (Figure 1B), and an autocrine loop in which CAFs send ligands which they also sense by expressing their receptors. In fact, CAFs were the cell type with the highest interaction scores. These features are found also in mouse breast cancer scRNA-seq data ¹⁶. In mice, as in human data, CAFs are the most interacting cell type and their autocrine interaction, as well as their interaction with macrophages, are among the strongest (Figure 1B, S1B).

To better understand the structure of the interactions among the cell types, we constructed a weighted and directed cell-network from the interaction matrix (Figure 1C, S1C; See Methods for details). This analysis revealed that the interaction network is hierarchical and mostly feedforward, with CAFs at the top sending signals to the other cell types, and TAMs at the bottom, receiving signals from CAFs and other cell types (Figure 1C).

We next asked whether the complex network of interactions can be simplified and described in terms of repeating instances of smaller circuits, which we can isolate and further explore. For this purpose we employed network motif analysis, which detects small circuits that occur in the network significantly more often than in randomized networks. We began with circuits made of two interacting cell types. Of the 7 possible connected two-cell circuits (Figure 1D), we found that only one type recurs with an interaction score that exceeds those found in randomized networks, and is thus a network motif ^{12,13}. This circuit has two interacting cell types that send mutual paracrine signals, and each also has an autocrine signaling loop. This circuit appears three times in the network, and all have CAFs as one of the nodes. Of these circuits the strongest circuit in terms of total and mean interaction score is the CAF-TAM circuit (Figure 1E).

We also analyzed circuits made of three and four cell types (Figure 1F-G, S1D). The most common circuits were made of combinations of the above-mentioned two-cell circuit (Figure 1E). The CAF-TAM pair participated in the highest scoring instances of these three- and four-cell circuits (Figure 1G, S1D). For example, the CAF-TAM circuit interacts with cancer cells to form a three-cell circuit in which CAFs send signals to cancer cells which in turn send signals to TAMs (Figure 1H).

We conclude that the breast cancer TME interaction network is hierarchical and composed of repeated occurrences of a specific two-cell circuit motif. CAFs are at the top of the hierarchy and send out many signals, whereas TAMs are at the bottom receiving end. The CAF-TAM two-cell circuit is an example of this motif and is one of the circuits with the strongest interactions in the network. Thus, studying this circuit in isolation can improve our understanding of the TME interaction network.”

Figure 5: RARRES2 and its receptor CMKLR1 are part of the CAF-TAM signaling axis in breast cancer. A. Venn diagram of potential CAF-to-TAM ligands shared between breast cancer patient tumors (Pal et al., 2021) and 4T1 tumors in mice (Sebastian et al., 2020a) based on the Nichnet ligand-receptor tool. **B.** Heatmap representation of the expression of the 77 shared ligands from (A) in fibroblasts from the *in vitro* cell-circuit, based on bulk RNA-seq data from Figure 4B. **C-D.** qPCR of *Rarres2* and *Cmk1r1* genes from fibroblasts and macrophages monocultured or co-cultured in control or cancer CM for 72 hours. **E.** CAFs and cancer cells from 4T1 tumors grown in syngeneic mice were isolated, cultured for 72h and RARRES2 levels in the media were assessed by ELISA. **F.** Transwell migration assay of macrophages in the presence of control medium, or 4T1 cancer CM with or without recombinant RARRES2 for 24h. Luminescence values were normalized to 4T1-CM in log₂. Data are combined from at least three independent

experiments, with a total of n=9-12 biological replicates. P-value was calculated using one-way ANOVA followed by Tukey's multiple comparisons test. Error bars represent SEM, *p < 0.05, **p < 0.01, ***p < 0.001, ****p < 0.0001. **G.** Violin plot of *RARRES2* expression based on human scRNA-seq of normal fibroblasts (N=17) versus CAFs (N=32). P-value was calculated using Students' T-test, ****p < 0.0001. **H.** *RARRES2* expression in different clusters from human scRNA-seq data, including CAF subclusters (defined using markers shown in Figure S5A). **I.** Pearson correlation between fibroblasts expressing high levels of *RARRES2* and macrophages expressing high levels of the protumorigenic TAM signature. Each dot represents one patient, based on human scRNA-seq (Pal et al., 2021). **J.** *RARRES2* gene expression in breast cancer patients (Pal et al., 2021) stratified by grade (high grade = grade 3, N=21; low-grade = grade 1 and 2, N=10). P-value was calculated using Students' T-test, **p < 0.01.

“The isolated circuit highlights RARRES2 and CMKLR1 as potential mediators of CAF-TAM signaling

The isolated cell circuit enables testing the effect of specific ligand-receptor interactions on the dynamics, function and transcriptomes of the circuit. We therefore asked which ligands are upregulated in both human and mouse CAFs, as well as in the co-culture. First, we identified shared ligands between human and mouse breast CAFs based on scRNA-seq datasets ^{14,16}, using the NicheNet tool ³². This analysis revealed 77 shared ligands which have potential receptors on TAMs (Figure 5A). Next, we assessed the expression of these ligands in the RNA-seq data from our co-culture (Figure 5B). We identified a cluster of 23 ligands upregulated in the presence of cancer CM (Figure 5B, cluster 1). We validated the expression of several genes from this cluster by qRT-PCR and found that indeed they were all significantly upregulated in fibroblasts co-cultured in the presence of cancer CM (Figure 5C, S5C).

*This cluster included genes well known to mediate CAF-TAM interactions in cancer, such as *Tgfb1*, *Csf1*, and *Ccl2* ³³. It also included Retinoic Acid Receptor Responder (*Rarres2*; also known as Chemerin), a chemokine known to regulate inflammation, adipogenesis, and metabolism through activation of the chemokine-like receptor 1 (*CMKLR1*) ^{34,35}. *RARRES2* was recently found to be expressed by CAFs in colorectal cancer ³⁶. However, its role in the TME was not elucidated. We therefore decided to focus our analyses on *RARRES2*. We found that *Rarres2* is upregulated in fibroblasts cultured in the presence of cancer CM, and its expression is further induced by co-culture with macrophages (Figure 5C). Strikingly, its receptor, *Cmklr1*, is upregulated in macrophages upon co-culture with fibroblasts or in the presence of cancer CM (Figure 5D). To directly test whether *RARRES2* is not only expressed by CAFs but also secreted by them, we isolated CAFs from 4T1 tumors, cultured them for 3 days and measured the levels of *RARRES2* in the medium by ELISA. We also isolated and cultured the 4T1 cancer cells themselves, as control (Figure 5E). This analysis confirmed that *RARRES2* is secreted from CAFs and not from cancer cells. To gain insight into how *RARRES2* affects TAMs, we monitored macrophage proliferation and migration in the presence of control medium, cancer CM, and cancer CM with recombinant *RARRES2*. *RARRES2* did not affect macrophage proliferation (Figure S5D). It did however significantly enhance macrophage migration (Figure 5F), highlighting the potential role of *RARRES2*-*CMKLR1* signaling in breast cancer.*

*To test the relevance of our findings to human disease, we compared *RARRES2* expression in CAFs to fibroblasts from normal breast. Analysis of the human breast scRNA-seq dataset ¹⁴ showed that *RARRES2* is specifically upregulated by CAFs and not by normal mammary fibroblasts (Figure 5G). Moreover, it is uniquely expressed by CAFs and not by other cell types within the human breast TME (Figure 5H, S5A). Next, to explore the potential effects of *RARRES2* expression on TAMs in patients, we computed the correlation between CAFs expressing high levels of *RARRES2* and macrophages expressing a protumorigenic TAM signature associated*

with poor survival ²⁹ in the human breast cancer RNA-seq dataset ¹⁴. We found a strong correlation, consistent with RARRES2 expression contributing to a protumoral TAM phenotype (Figure 5I). To further assess the clinical relevance of RARRES2 expression, we compared the expression levels of RARRES2 in low-grade vs high-grade tumors. We found that RARRES2 expression is significantly higher in high-grade cases compared to low-grade cases (Figure 5J). We conclude that RARRES2, identified through our in-vitro circuit approach, mediates CAF signaling to TAMs in mouse models of breast cancer and in human disease, and may serve as a therapeutic target for future exploration.”

To improve the quality of the article, the authors need to address the following questions.

(1) The authors measured cell counts for 3d and 7d, and later for 14d and 21d. I wonder why the authors didn't use the cell counts for the initial conditions (0d). I asked this because even for 3d the cell count data were clustered in narrowed regions in the phase plane (counts of fibroblasts vs. macrophages). My guess is that the initial conditions were widespread, but because of the temporal dynamics during 0 – 3 days, the tested conditions at 3d are much narrowed. Thus, cell counts for many regions were not accessible anymore at 3d.

Response: We thank the reviewer for the positive assessment of our results and helpful suggestions. We now clarify why we did not use time point zero:

Lines 648-655: “We chose day 3 as the initial time point (and not an earlier time point such as day 0), to ensure the cells are settled in the 2D layer in terms of secreted factor interactions. During the first day after seeding interactions occur in 3D within the entire well volume, since it takes time for the cells to settle and form a 2D layer at the bottom. Thus, their effective density during this phase is very low and below the separatrix, predicting that their numbers should crash. Indeed, we seeded cells with varied initial conditions spanning the entire range of the phase plane and observed that cell numbers at day 3 are much smaller than at seeding.”

(2) In Figure 1B-G, how were the steady states defined by the vectors? It seems that the states were derived from Figure 2C-F, although it was not mentioned in Figure 1, and not all the states from Figure 2 were listed in Figure 1. The authors need to clarify if the states were obtained from model fitting, or they were defined directly by the vectors. Because of the sparsity of the “vector field”, to me it is hard to determine the separatrix and in turn the unstable/saddle steady states. In fact, some vectors in lung/fat/cancer conditions were clearly opposite to the directions that were supposed to be defined by the assigned steady states. The authors have already provided the goodness of fitting by plotting observed and predicted cell counts in day 7 (in the logscale though) in Figure S3 I-P. It would be important to also evaluate the direction of the changes from observation and prediction.

Response: This comment helped us to clarify our approach. We used an intuitive and a precise approach. Intuitively we detect stable fixed points as points to which arrows converge from all directions, and semi stable points as those to which arrows converge only from two sides and not from the other two. Their precise position on the plot was determined by the model fit, as we note in figure 2 legend. The precise position of the steady states were indeed defined by model fitting.

We now also estimate the model performance in terms of error in the predicted direction of the arrows (see Figure S3E-F): The great majority of the predicted arrows point correctly to a median error of 16° in control and 22° in cancer CM.

Lines 273-276: “The model showed good fits for the experimental dynamics, explaining 84%-93% of the variance in the data (Figure S3A-D) with a reasonably low error in predicting the direction of growth of the cell populations (Figure S3E-F).”

Supplementary Figure 3: Evaluation and validation of the fibroblast-macrophage circuit mathematical model.

A-D. Predicted cell numbers at day 7 were plotted against the observed numbers to assess the goodness of fit of the model in control medium (**A, C**) and in cancer CM (**B, D**). Dashed lines indicate perfect fit. **E-F.** The error in prediction of the direction of growth of the cell populations: The error between the direction of the observed arrow to the direction of the predicted arrow is shown.

(3) In page 8, the authors mentioned “standard information criteria”. It is beneficial to refer to the AIC methods from the Method section.

Response: We removed model selection from the manuscript since it does not add to the main conclusions. Instead we infer the parameters from a single model that includes all the parameters. This preserves the conclusions - hierarchy of strengths with the F autocrine loop strongest - while increasing the rigor of the analysis.

(4) In the results of Figure 3O-R and Figure S3 A-H, the authors evaluated AIC of the full model and the models with one less parameter. I wonder why the elimination of parameter is only limited by one. Also, it’s hard to determine the best model for cases when the multiple AIC scores are close to the minimum (e.g., Figure S3A, Full and -P_FM). Would the parameter fitting work well when P_FM is also eliminated for mammary? In addition, I believe the labels P_FM

and P_MF were swapped in Figure S3A-H (I used in the original labels in the previous sentences).

Response: We thank the reviewer for this comment. As mentioned in our response to the previous comment (3), we no longer use model selection/AIC. This removes the need for model selection criteria.

(5) In Figure 3AEF, the authors show nicely the pathways for different gene clusters from macrophages. I suggest the authors also do so for gene clusters from fibroblast (Figure S5B).

Response: We thank the reviewer for this suggestion. We now focus on the mammary circuit, with and without cancer conditioned media, and added an analysis of the differentially regulated gene clusters and pathways in both macrophages and fibroblasts in these conditions (Figure 4).

Figure 4: RNA sequencing supports predicted changes in macrophage and fibroblast cell circuits in cancer-conditioned medium. A-D. Heatmaps showing hierarchical clustering of differentially expressed genes (DEGs; basemean > 5; |LogFoldChange| > 1; FDR < 0.1). **A.** An interaction model (medium and culture) was used to compare DEGs between macrophages mono-cultured, or co-cultured with mammary fibroblasts in DMEM vs cancer CM. The mono-cultured macrophages in DMEM were collected at day 0 (since they cannot maintain themselves in DMEM for 7 days), and at day 7 in cancer CM. The co-cultured macrophages were collected after 7 days of co-culture with mammary fibroblasts, in either DMEM or cancer CM. Macrophages in DMEM: n=3, macrophages in cancer CM: n=4 mice. **B.** An interaction model (medium and culture) was used to compare DEGs between fibroblasts mono-cultured (only), or co-cultured, with macrophages in cancer CM vs. DMEM. Fibroblasts in DMEM n=3, Fibroblasts in cancer CM: n=2-4 mice. **C-D.** Pathway analysis of the macrophage clusters from (A) and fibroblast clusters from (B) was conducted using

Metascape (Y. Zhou et al., 2019). Selected significant pathways are shown, see full list in Supplementary Table 3-4 (P < 0.05; FDR < 0.05).

(6) In Figure 4AB, ICELLNET ligand-receptor analysis was applied to RNA-seq data for the mammary circuits. Could this be performed for the other two organs? I wonder if the M to F interactions were significantly weaker in these cases.

Response: The analysis could be performed on normal fat and lung fibroblasts, however since our revised manuscript focuses mostly on the mammary cell-circuit in the context of cancer we focus also our ligand-receptor interaction analysis on potential CAF ligands and TAM receptors, and use for this scRNA-seq data from human and mouse breast cancer (Figure 5).

Figure 5: RARRES2 and its receptor CMKLR1 are part of the CAF-TAM signaling axis in breast cancer. **A.** Venn diagram of potential CAF-to-TAM ligands shared between breast cancer patient tumors (Pal et al., 2021) and 4T1 tumors in mice (Sebastian et al., 2020a) based on the Nichet ligand-receptor tool. **B.** Heatmap representation of the expression of the 77 shared ligands from (A) in fibroblasts from the *in vitro* cell-circuit, based on bulk RNA-seq data from Figure 4B. **C-D.** qPCR of *Rarres2* and *Cmk1r1* genes from fibroblasts and macrophages monocultured or co-cultured in control or cancer CM for 72 hours. **E.** CAFs and cancer cells from 4T1 tumors grown in syngeneic mice were isolated, cultured for 72h and RARRES2 levels in the media were assessed by ELISA. **F.** Transwell migration assay of macrophages in the presence of control medium, or 4T1 cancer CM with or without recombinant RARRES2 for 24h. Luminescence values were normalized to 4T1-CM in log₂. Data are combined from at least three independent

experiments, with a total of n=9-12 biological replicates. P-value was calculated using one-way ANOVA followed by Tukey's multiple comparisons test. Error bars represent SEM, *p < 0.05, **p < 0.01, ***p < 0.001, ****p < 0.0001. **G.** Violin plot of *RARRES2* expression based on human scRNA-seq of normal fibroblasts (N=17) versus CAFs (N=32). P-value was calculated using Students' T-test, ****p < 0.0001. **H.** *RARRES2* expression in different clusters from human scRNA-seq data, including CAF subclusters (defined using markers shown in Figure S5A). **I.** Pearson correlation between fibroblasts expressing high levels of *RARRES2* and macrophages expressing high levels of the pro-tumorigenic TAM signature. Each dot represents one patient, based on human scRNA-seq (Pal et al., 2021). **J.** *RARRES2* gene expression in breast cancer patients (Pal et al., 2021) stratified by grade (high grade = grade 3, N=21; low-grade = grade 1 and 2, N=10). P-value was calculated using Students' T-test, **p < 0.01.

(7) In page 22 the third paragraph, it seems that Figure S6G was wrongly cited.

Response: Thank you, this section has been modified.

(8) In the modeling equations (2-3) in page 28, if $f(X)$ is chosen to be the logarithm function (btw, is that \log_{10} ?), then K_F and K_M would have different meanings than carrying capacity. The terms have been used in log scale and non-logscale in multiple places (e.g. throughout Figure 2).

Response: Choosing $f(X)$ to be logarithmic means that the carrying capacity parameters are also represented in logarithmic scale. We now note that the base of the logarithm in the equations is e (natural logarithm). In the text and Figures, we converted the natural log in some cases to base 10 for clarity and more intuitive understanding.

We now address these points in the Methods section:

Lines 883-891: "Exploration of the data favored $f(X) = \log(X + 1)$, where \log is the natural logarithm, which represents a nonlinear relationship with diminishing relative effects of large cell populations. This nonlinear relationship provided better fits than a linear one, $f(X) = X$ (Figure S6) Adding 1 inside the log is common when working with counts, in order to avoid infinity at zero cells. The function f resembles the saturation effect in Michaelis-Menten (MM) interactions. We chose not to use MM expressions to keep the lowest number of parameters possible, because each MM term requires an additional 'halfway point' parameter. Using log means that carrying capacities are in log cell numbers."

(9) Related to the previous question: the authors mentioned "This nonlinear relationship provides better fits than a linear one". Could the authors provide quantitative evidence?

Response: We now add to the Supplementary Information a Figure concerning this point (Figure S6). Fitting the model with a linear function, $f(X)=X$, zeros out all the rate parameters (autocrine and paracrine interactions and removal rates) for macrophages and for fibroblasts in both conditions. Intuitively, in this case the model simply predicts that the cell number in day 7 is the cell number in day 3. Our model, with the non-linear relationship, performs better and is more biologically relevant (because we know that ligand-receptor interactions saturate) than the linear model.

Lines 885-886: "This nonlinear relationship provided better fits than a linear one, $f(X) = X$ (Figure S6)"

Figure S6: Evaluation of a mathematical model with a linear dependence on cell population size $f(X) = X$. A-D. Goodness of fit by plotting predicted vs. observed number of cells on day 7. **E-F.** The error in prediction of the direction of growth of the cell populations: The error between the direction of the observed arrow to the direction of the predicted arrow is shown.

(10) In data availability, the bulk RNAseq data was described as deposited, but it is still not accessible. Also, the authors need to deposit the cell cytometry data, at least the cell counts table for various conditions.

Response: All data will be deposited as required.

Reviewer #3 (Remarks to the Author): with expertise in mathematical biology

The authors present an integrative work regarding the identification of macrophages and fibroblasts interactions in the tumor microenvironment. The work involves in vitro experiments with murine cells where co-cultures of fibroblasts and macrophages were embedded in normal and cancer microenvironments. In turn, a mathematical model was fitted to experiments. Finally, transcriptional profiling of CAFs taken from mice and humans show similar traits to the ones identified in the in vitro experiments. The work is very interesting but before publication I would like the authors to address the following points:

Response: We thank the reviewer for this endorsement.

- The in vitro model reflects the situation where resident macrophages interact with fibroblasts. However, it is known that in vivo most of macrophages are recruited from the blood stream. It would be interesting to know how this would impact the system. Certainly, this could be addressed in the mathematical model.

Response: We agree, we would like to clarify that we utilized bone-marrow-derived macrophages (BMDMs) in our experiments. These cells are considered to be the main source of macrophages recruited from the bloodstream into tumors (Shi, C., Pamer, E. 2011).

- Regarding the parameter calibration I was wondering if this parameter value selection is a local minimum. Does the algorithm perform any perturbation to identify the robustness of the parameter selections? Another interesting point would be to analyze the sloppiness (literature from J. Senth) of the system and find out which parameters impact the system behavior the most. Alternatively, one could make bifurcation diagrams from all parameters, since they are not so many.

Response: We thank the reviewer for this suggestion. We now validate our model inference with Bayesian tools to show convergence of parameters guessed from different initial conditions. Thus, indicating a more robust parameter inference.

Appendix 1: Validation of parameter calibration with Bayesian inference implemented in the python package PyDREAM (Shockley et al., 2018). Trace plots of five independent Markov chains for each condition (control at the top two rows and cancer at the bottom two) with zoom (insets) on the last 2,000 samples to show convergence.

Appendix 2: Parameter distributions, pooled from all 2,000 last samples of the five Markov chains. The parameter distributions are similar to the ones in Figure 3C obtained by bootstrapping.

- I know that the authors didn't study the impact of other determinants such as hypoxia or inflammation (and the consequent polarization of the macrophages) but they could discuss it maybe in the light of a recent paper Setten et al, Nat. Comms, 2022.

Response: We now added discussion of determinants such as hypoxia and inflammation, according to the important recent advance by Setten et. al (2022).

Lines 565-568: "Recent work in the context of kidney fibrosis has demonstrated how local inflammation and hypoxia fields can affect a circuit's parameters⁹, and thus the same circuit may have different steady-states in different parts of the tissue."

- Finally a paragraph with the study limitations would be very useful.

Response: Thank you, we added this:

Lines 565-570: "One limitation of the present approach is the lack of spatial effects in the analysis. Recent work in the context of kidney fibrosis has demonstrated how local inflammation and hypoxia fields can affect a circuit's parameters⁹, and thus the same circuit may have different steady-states in different parts of the tissue. Emerging spatial omic approaches can help to reveal the effects of such spatial fields on the circuits, and to form a quantitative understanding of the heterogeneous tumor landscape."

Reviewer #4 (Remarks to the Author): with expertise in fibroblasts/macrophages cross-talk, cell-to-cell communication

Cell circuits and fibroblasts and macrophages in the tumour microenvironment.

Here the authors measure population dynamics of directly co-cultured macrophages and fibroblasts to identify regulators of cell circuits. The authors observe that whereas monocultured fibroblasts are self-sustained, macrophages are not and rely on interactions with co-cultured fibroblasts. Moreover, and curiously, the authors identify minor differences in the interactions between macrophages and fibroblasts from breast, mesometrium and the lung. More significantly, tumour cell conditioned medium offset the population dynamics. Finally, the authors use transcriptional and computational analysis to identify putative signals regulating the cell circuits.

Overall, the manuscript is nicely presented and the work is of high quality.

Response: We thank the reviewer for this endorsement.

However, the conceptual advances are minor and there are limited direct molecular evidence included to substantiate the authors' claims.

Response: We thank the reviewer for providing helpful comments. We have made significant additions to our manuscript that enhance the novelty of our study. Specifically, we included a circuit motif analysis of the network of all interactions between different cell types in the tumor microenvironment. Surprisingly, we found that one circuit motif was far more prevalent than others and appeared repeatedly with different cell types. This circuit motif includes an autocrine loop of two cell types and bi-directional paracrine interactions with another cell type, which precisely matches the motif observed in our in-vitro model. Of all possible options, the most abundant instance of this motif is the CAF-TAM interaction. The new network motif analysis therefore provides a framework and rationale for focusing on CAF-TAM interactions (Figure 1, see below with the related text, lines 77-120).

Our revised manuscript also more clearly emphasizes the conceptual advance over earlier work. We offer a new concept for simplifying the tumor microenvironment into small circuits of a few cell types which can be studied in-vitro, while retaining their in-vivo relevance. This allows characterizing novel interaction molecules, and understanding the dynamics and bistability properties of the TME at unprecedented quantitative resolution.

We further use the network analysis and circuit approach to identify a novel CAF-TAM interaction between RARRES2 and its receptor CMKLR1. We performed new experiments and analyses that provide direct molecular evidence supporting the notion that RARRES2 enhances macrophage migration, and has clinical relevance in human data (Figure 5, see below with the related text, lines 419-464). Overall, these findings extend far beyond what we originally anticipated and provide novel insights into the biology of the tumor microenvironment.

Figure 1: Network analysis reveals a hierarchy of interactions with a dominant CAF-TAM circuit in the breast tumor microenvironment. **A.** UMAP visualization of the main cell clusters in the breast TME in human scRNA-seq data from 32 patients ¹⁴. **B.** Heatmap of interaction strengths between pairs of cells based on cumulative ligand-receptor interaction scores using CellChat ¹⁵ applied to the scRNA-seq data of **(A)**. **C.** Illustration of the structure of the network based on the analysis in **(B)** shows hierarchy (see Methods). Arrow width is proportional to the interaction score. **D.** Network motif analysis of all 7 possible two-cell circuit patterns, tested for abundance relative to randomized degree-preserving networks. The dashed line represents a 0.05 p-value threshold. **E.** The top three two-cell interaction subgraphs scored by the average weight all include CAFs. **F.** Network motif analysis of the 13 possible three-cell-circuit patterns. The dashed line represents a 0.05 p-value threshold. **G.** The top three scoring three-cell circuit subgraphs. **H.** The three-cell circuit of CAFs, TAMs and cancer cells.

“Analysis of the breast cancer microenvironment reveals a hierarchy of interactions with a dominant CAF-TAM circuit

To begin to untangle the complexity of the TME, we mapped the network of cell-cell interactions by analyzing published scRNA-seq data from breast cancer patients ¹⁴. We identified the cell types (Figure 1A and S1A) and scored their interactions (Supplementary Table 1) using CellChat ¹⁵, a tool for estimating ligand-receptor interaction strength. We found multiple pairs of interacting cell types (Figure 1B), and used CellChat to score the strength of each interaction.

We found that the strongest interaction occurred between CAFs and myeloid cells (comprised mainly of tumor-associated macrophages, hereafter referred to as TAMs), followed by the interaction between CAFs and mast cells (Figure 1B), and an autocrine loop in which CAFs send ligands which they also sense by expressing their receptors. In fact, CAFs were the cell type with the highest interaction scores. These features are found also in mouse breast cancer scRNA-seq data ¹⁶. In mice, as in human data, CAFs are the most interacting cell type and their autocrine interaction, as well as their interaction with macrophages, are among the strongest (Figure 1B, S1B).

To better understand the structure of the interactions among the cell types, we constructed a weighted and directed cell-network from the interaction matrix (Figure 1C, S1C; See Methods for details). This analysis revealed that the interaction network is hierarchical and mostly feedforward, with CAFs at the top sending signals to the other cell types, and TAMs at the bottom, receiving signals from CAFs and other cell types (Figure 1C).

We next asked whether the complex network of interactions can be simplified and described in terms of repeating instances of smaller circuits, which we can isolate and further explore. For this purpose we employed network motif analysis, which detects small circuits that occur in the network significantly more often than in randomized networks. We began with circuits made of two interacting cell types. Of the 7 possible connected two-cell circuits (Figure 1D), we found that only one type recurs with an interaction score that exceeds those found in randomized networks, and is thus a network motif ^{12,13}. This circuit has two interacting cell types that send mutual paracrine signals, and each also has an autocrine signaling loop. This circuit appears three times in the network, and all have CAFs as one of the nodes. Of these circuits the strongest circuit in terms of total and mean interaction score is the CAF-TAM circuit (Figure 1E).

We also analyzed circuits made of three and four cell types (Figure 1F-G, S1D). The most common circuits were made of combinations of the above-mentioned two-cell circuit (Figure 1E). The CAF-TAM pair participated in the highest scoring instances of these three- and four-cell circuits (Figure 1G, S1D). For example, the CAF-TAM circuit interacts with cancer cells to form a three-cell circuit in which CAFs send signals to cancer cells which in turn send signals to TAMs (Figure 1H).

We conclude that the breast cancer TME interaction network is hierarchical and composed of repeated occurrences of a specific two-cell circuit motif. CAFs are at the top of the hierarchy and send out many signals, whereas TAMs are at the bottom receiving end. The CAF-TAM two-cell circuit is an example of this motif and is one of the circuits with the strongest interactions in the network. Thus, studying this circuit in isolation can improve our understanding of the TME interaction network.”

Figure 5: RARRES2 and its receptor CMKLR1 are part of the CAF-TAM signaling axis in breast cancer. **A.** Venn diagram of potential CAF-to-TAM ligands shared between breast cancer patient tumors (Pal et al. 2021) and 4T1 tumors in mice (Sebastian et al. 2020) based on the Nichnet ligand-receptor tool. **B.** Heatmap representation of the expression of the 77 shared ligands from (A) in fibroblasts from the *in vitro* cell-circuit, based on bulk RNA-seq data from Figure 4B. **C-D.** qPCR of *Rarres2* and *Cmk1r1* genes from fibroblasts and macrophages monocultured or co-cultured in control or cancer CM for 72 hours. **E.** CAFs and cancer cells from 4T1 tumors grown in syngeneic mice were isolated, cultured for 72h and RARRES2 levels in the media were assessed by ELISA. **F.** Transwell migration assay of macrophages in

the presence of control medium, or 4T1 cancer CM with or without recombinant RARRES2 for 24h. Luminescence values were normalized to 4T1-CM in log2. Data are combined from at least three independent experiments, with a total of n=9-12 biological replicates. P-value was calculated using one-way ANOVA followed by Tukey's multiple comparisons test. Error bars represent SEM, *p < 0.05, **p < 0.01, ***p < 0.001, ****p < 0.0001. **G.** Violin plot of RARRES2 expression based on human scRNA-seq of normal fibroblasts (N=17) versus CAFs (N=32). P-value was calculated using Students' T-test, ****p < 0.0001. **H.** RARRES2 expression in different clusters from human scRNA-seq data, including CAF subclusters (defined using markers shown in Figure S5A). **I.** Pearson correlation between fibroblasts expressing high levels of RARRES2 and macrophages expressing high levels of the pro-tumorigenic TAM signature. Each dot represents one patient, based on human scRNA-seq (Pal et al., 2021). **J.** RARRES2 gene expression in breast cancer patients (Pal et al., 2021) stratified by grade (high grade = grade 3, N=21; low-grade = grade 1 and 2, N=10). P-value was calculated using Students' T-test, **p < 0.01.

“The isolated circuit highlights RARRES2 and CMKLR1 as potential mediators of CAF-TAM signaling

The isolated cell circuit enables testing the effect of specific ligand-receptor interactions on the dynamics, function and transcriptomes of the circuit. We therefore asked which ligands are upregulated in both human and mouse CAFs, as well as in the co-culture. First, we identified shared ligands between human and mouse breast CAFs based on scRNA-seq datasets ^{14,16}, using the NicheNet tool ³². This analysis revealed 77 shared ligands which have potential receptors on TAMs (Figure 5A). Next, we assessed the expression of these ligands in the RNA-seq data from our co-culture (Figure 5B). We identified a cluster of 23 ligands upregulated in the presence of cancer CM (Figure 5B, cluster 1). We validated the expression of several genes from this cluster by qRT-PCR and found that indeed they were all significantly upregulated in fibroblasts co-cultured in the presence of cancer CM (Figure 5C, S5C).

This cluster included genes well known to mediate CAF-TAM interactions in cancer, such as Tgfb1, Csf1, and Ccl2 ³³. It also included Retinoic Acid Receptor Responder (Rarres2; also known as Chemerin), a chemokine known to regulate inflammation, adipogenesis, and metabolism through activation of the chemokine-like receptor 1 (CMKLR1) ^{34,35}. RARRES2 was recently found to be expressed by CAFs in colorectal cancer ³⁶. However, its role in the TME was not elucidated. We therefore decided to focus our analyses on RARRES2. We found that Rarres2 is upregulated in fibroblasts cultured in the presence of cancer CM, and its expression is further induced by co-culture with macrophages (Figure 5C). Strikingly, its receptor, Cmk1r1, is upregulated in macrophages upon co-culture with fibroblasts or in the presence of cancer CM (Figure 5D). To directly test whether RARRES2 is not only expressed by CAFs but also secreted by them, we isolated CAFs from 4T1 tumors, cultured them for 3 days and measured the levels of RARRES2 in the medium by ELISA. We also isolated and cultured the 4T1 cancer cells themselves, as control (Figure 5E). This analysis confirmed that RARRES2 is secreted from CAFs and not from cancer cells. To gain insight into how RARRES2 affects TAMs, we monitored macrophage proliferation and migration in the presence of control medium, cancer CM, and cancer CM with recombinant RARRES2. RARRES2 did not affect macrophage proliferation (Figure S5D). It did however significantly enhance macrophage migration (Figure 5F), highlighting the potential role of RARRES2-CMKLR1 signaling in breast cancer.

To test the relevance of our findings to human disease, we compared RARRES2 expression in CAFs to fibroblasts from normal breast. Analysis of the human breast scRNA-seq dataset ¹⁴ showed that RARRES2 is specifically upregulated by CAFs and not by normal mammary fibroblasts (Figure 5G). Moreover, it is uniquely expressed by CAFs and not by other cell types within the human breast TME (Figure 5H, S5A). Next, to explore the potential effects of RARRES2 expression on TAMs in patients, we computed the correlation between CAFs expressing high

levels of RARRES2 and macrophages expressing a protumorigenic TAM signature associated with poor survival ²⁹ in the human breast cancer RNA-seq dataset ¹⁴. We found a strong correlation, consistent with RARRES2 expression contributing to a protumoral TAM phenotype (Figure 5I). To further assess the clinical relevance of RARRES2 expression, we compared the expression levels of RARRES2 in low-grade vs high-grade tumors. We found that RARRES2 expression is significantly higher in high-grade cases compared to low-grade cases (Figure 5J). We conclude that RARRES2, identified through our in-vitro circuit approach, mediates CAF signaling to TAMs in mouse models of breast cancer and in human disease, and may serve as a therapeutic target for future exploration.”

Specifically:

1. The authors describe how fibroblasts and macrophages differ in their interdependencies. However, these experiments are done in the presence of 10%FBS, which acts as a significant supporter of fibroblasts but perhaps not as much for macrophages. The point being that the experimental setup may be biased in support of their observations and consequently raises concerns about the meaningfulness of these.

Similarly, the authors use conditioned medium from a cancer cell line and claim this disrupts the observed cell circuitries. However no normal control is used for comparison. Moreover, tumour cells exist within a cellular complex environment with a resulting change to the tumour cell transcriptional phenotype e.g. tumour cell conditioned medium does not take into account the effects of host cells on tumour cell gene expression. Finally, there is also a distinct possibility the observed interactions are due to differences in metabolic requirements, but this is not even discussed.

Response: We thank the reviewer for these comments. To address the concern regarding cancer CM, we performed a new experiment to test the proliferation of macrophages in the presence of CM from normal (non-malignant) epithelial cells. We found that CM from the non-malignant epithelial cells indeed had a growth-promoting effect, however the effect of cancer CM was significantly larger (Figure 2H, see below).

Figure 2H. Macrophage cell numbers counted using a cell titer glo kit, following three days of growth in mono-culture in the presence of DMEM or in the presence of CM from normal mouse epithelial cells or from 4T1 cancer cells. Data are combined from three independent experiments, with n=8 biological replicates. P-value was calculated using one way ANOVA. Error bars represent SEM, **p < 0.001 ****p < 0.0001.

Lines 209-215: “Macrophages in cancer CM were able to grow in the absence of fibroblasts (red arrows; Figure 2F), in contrast to control media, in which their growth depends on fibroblasts (gray arrows; Figure 2F). This may relate to the composition of 4T1-conditioned medium which contains factors that regulate macrophage proliferation 23. A growth-promoting effect was also observed when macrophages were grown in CM from normal (non-malignant) mammary epithelial cells, however this effect was mild compared to the growth-promoting effect of cancer CM (Figure 2H).”

Concerning the first point, the dependence of fibroblasts on FBS was previously tested by Zhou et al., 2018, who showed that macrophage to fibroblast ratios are similar in co-cultures regardless of serum concentration (Figure S4C in Zhou et al. 2018).

We agree with the reviewer that fibroblasts grow better than most other cell types in culture. Interestingly, our new network motif analysis shows that this actually resembles the in-vivo situation where fibroblasts are less dependent on signals from other cell types than the rest of the cell types (Figure 1C, see below).

As a technical note we would also like to add that we used standard growth conditions including not only FBS but also nutrients and glucose in full growth medium (DMEM). These growth conditions are used routinely for most cell cultures.

Figure 1C. Illustration of the structure of the network based on the analysis in (B) shows hierarchy (see Methods). Arrow width is proportional to the interaction score.

2. The authors present some very interesting quantitative projections of the cell circuitries to identify differences between interactions between macrophages and fibroblasts from different origins. However, it is difficult to estimate whether small difference between numbers e.g. 0 and 0.02 (Fig 2o-r) are functionally meaningful without quantitative validation.

Response: We agree, and now bypass this issue by avoiding model selection that compares models with strict 0 parameter values to models with fitted parameters. Instead we now use a single model and estimate all parameters without restricting them to be zero or nonzero.

3. The authors use RNA expression analysis to identify putative ligand-receptor pairs that may mediate the differences in cell interactions. Firstly, there is no functional analysis to confirm that indeed differences in paracrine cell signalling is a major contributor compared to other changes such as metabolic capacity. Moreover, the functional relevance of the predicted interactions are not validated and thus it remains unclear whether the observed expression changes are related to the described cell circuitries.

Response: This comment helped us to add computational and functional analyses to support our findings, and to characterize a new paracrine factor that has clinical relevance in human data. We performed a ligand-receptor interaction analysis based on two scRNA-seq datasets of human breast cancer patients and the mouse 4T1 model. We identified shared upregulated ligands in CAFs that have potential receptors in TAMs. We then used this list to analyze our RNA-seq data and identified 23 potential ligands.

In our analysis, we focused on RARRES2 and conducted several experimental validations on its secretion from CAFs and its influence on macrophage migration and promotion of protumoral profile. Additionally, we found that RARRES2 expression is associated with higher disease grade (Figure 5, see above). These additional analyses support our conclusions and provide functional evidence for the role of CAF-TAM interactions in promoting tumor progression.

4. Moreover, the authors note that macrophage gene expression is more strongly affected by fibroblast co-culture than fibroblasts by macrophage co-culture. This could also be a reflection of distinct sensitivities to the growth conditions eg DMEM/FBS impacts fibroblast gene expression to a larger extent than it affect macrophage gene expression and thus fibroblasts appear less sensitive to the co-cultures that macrophages.

Response: This comment helped us clarify the roles of conditioned-medium. We emphasize that in the cancer CM where both cells are in their optimal growth condition, we found that fibroblasts were not affected by the macrophages. However, macrophages in the presence of cancer CM and fibroblasts upregulated a unique cluster of genes that were not detected in macrophages alone. This finding provides additional evidence for macrophages being affected by fibroblasts, which is supported by our hierarchical network plot that also shows fibroblasts as the main sender signal and macrophages as the main receiver signal (based on scRNA-seq data from patient samples). Furthermore, we performed new experiments to test the proliferation of macrophages in the presence of normal epithelial CM and found that macrophage proliferation is dependent on secreted factors from cancer cells, as described in our response to comment 1 above.

REVIEWER COMMENTS

Reviewer #1 (Remarks to the Author):

I was very pleased to read this updated version of the manuscript. It is delightful to see that the authors took reviewer comments to heart and to see that the manuscript greatly improved. The authors have addressed all my concerns.

My only recommendation is that the authors include the figure labeled "Appendix 1" in the reviewer response as part of their manuscript (I assume in the supplement?) and mention this confirmation in the appropriate area of the paper. I think readers will find this useful and insightful.

I do not need to see the manuscript further.

Carlos F Lopez

Altos Labs

Reviewer #2 (Remarks to the Author):

In this revision, the authors had slightly shifted their focus on studying the effects of cancer on the fibroblast-macrophage circuit in breast cancer. They have included a new cell-cell interaction network analysis using scRNA-seq data, from which they identified the CAF-TAM circuit to be most significant cell circuit motif. This finding strengthens the motivation of this study. They also experimentally validate RARRES2 and CMKLR1 as a novel CAF-TAM interaction. Besides, they have properly addressed my questions from the previous review. Thus, I recommend publication of this manuscript in Nature communication.

Reviewer #3 (Remarks to the Author):

I think the authors answered to my concerns and I suggest the publication of the paper.

Reviewer #4 (Remarks to the Author):

The authors have a detailed response to the concerns initially raised, which has improved the quality of the manuscript. I cannot comment much more on the mathematical modelling, as done by the other reviewers, but I do have a couple of remaining queries for the authors to address. None of these should require substantial experimentation and most could be addressed by additional analysis and by refining the discussion.

The in vitro models are based on fibroblasts and macrophages isolated from normal healthy tissue, whereas the analysis of scRNAseq data focuses on tumour bearing tissue.

The reference included to the scRNA analysis of human breast cancer also includes normal and preneoplastic tissue, which should be analysed and compared to the in vitro model of normal tissue fibroblasts and macrophages (without cancer cell conditioned medium).

It is surprising, at least it is to me, that CAFs and not tumour cells emerge at the top of the network and that tumour cells are not observed as part of the network motif analysis. This is despite the fundamental requirement of tumour cells in establishing a TME (as it would be inflammation/fibrosis otherwise)? This is further substantiated by the observation that tumour cell conditioned medium bypass effects of CAFs on macrophage proliferation in vitro, which would be assumed to better represent the disease setting.

Thus, 1) Does the network structure change between normal and tumour bearing tissue as analysed by scRNAseq (the dataset used also includes normal healthy tissue)? For example, are the motifs different with different cell composition?

2) How is this reflected in the in vitro assay e.g. do the authors observe differences in their in vitro model +/- tumour cell conditioned medium that is reflected in the network analysis between healthy and tumour bearing tissue? This is of relevance in establishing whether the model reflects the in vivo condition, as hypothesised by the authors.

Moreover, the comparison between the network and structure with the in vitro data raise an interesting (and maybe important) question: Does the network structure/motifs or specific interactions/signals inform most critical elements of a normal vs diseased tissue? Specifically, tumour cell conditioned medium bypass macrophage dependence on

fibroblasts but CAFs are at the top of the in vivo network structure (in Fig 1). Does this imply that specific tumour cell signals are critical and dominant over their position and connectivity within the network?

I remain unconvinced that use of tumour cell conditioned medium fully replicates the interactions observed in a more complex system e.g. conscripted CAFs and TAMs cannot feed back to the tumour cells and thus the system would be anticipated to only replicate a tumour to fibroblast/macrophage signalling axis. Could the authors either include data to clarify this point or simply acknowledge this possible limitation in the discussion. Similarly, the authors denote fibroblasts as CAF and TAMs in their in vitro system, but in effect only replicate either normal or tumour-educated cells in vitro. The nomenclature should be clarified to avoid inconsistencies and misunderstandings.

The authors have included a measurement of collagen deposition (Fig 2E) and are concluding the increase in production reflects increased fibroblast ECM production. However, isn't the collagen deposition simply a readout of increased numbers of fibroblasts rather than a measure of their function given the increase in fibroblast proliferation as macrophage numbers are increased?

In Figure 4g the authors use ssGSEA to annotate CAF subsets, but I cannot see where they obtain the gene sets for the enrichment from – also, there is a lack of positive and negative controls for various CAF subtypes in this experiment.

Point-by-point response to the reviewers' comments

Reviewer #1 (Remarks to the Author):

I was very pleased to read this updated version of the manuscript. It is delightful to see that the authors took reviewer comments to heart and to see that the manuscript greatly improved. The authors have addressed all my concerns.

My only recommendation is that the authors include the figure labeled "Appendix 1" in the reviewer response as part of their manuscript (I assume in the supplement?) and mention this confirmation in the appropriate area of the paper. I think readers will find this useful and insightful.

I do not need to see the manuscript further.

Response: We appreciate this constructive feedback and support. We have now added Appendix 1 as Supplementary Figure 4 and added the following sentence to the main text: "We also validated convergence and robustness of parameter calibration by the Bayesian tool, PyDREAM²⁶ (Figure S4)."

Reviewer #2 (Remarks to the Author):

In this revision, the authors had slightly shifted their focus on studying the effects of cancer on the fibroblast-macrophage circuit in breast cancer. They have included a new cell-cell interaction network analysis using scRNA-seq data, from which they identified the CAF-TAM circuit to be most significant cell circuit motif. This finding strengthens the motivation of this study. They also experimentally validate RARRES2 and CMKLR1 as a novel CAF-TAM interaction. Besides, they have properly addressed my questions from the previous review. Thus, I recommend publication of this manuscript in Nature communication.

Response: We thank the reviewer for this endorsement.

Reviewer #3 (Remarks to the Author):

I think the authors answered to my concerns and I suggest the publication of the paper.

Response: We thank the reviewer for this endorsement.

Reviewer #4 (Remarks to the Author):

The authors have a detailed response to the concerns initially raised, which has improved the quality of the manuscript. I cannot comment much more on the mathematical modelling, as done by the other reviewers, but I do have a couple of remaining queries for the authors to address. None of these should require substantial experimentation and most could be addressed by additional analysis and by refining the discussion.

The in vitro models are based on fibroblasts and macrophages isolated from normal healthy tissue, whereas the analysis of scRNAseq data focuses on tumour bearing tissue. The reference included to the scRNA analysis of human breast cancer also includes normal and preneoplastic tissue, which should be analysed and compared to the in vitro model of normal tissue fibroblasts and macrophages (without cancer cell conditioned medium).

It is surprising, at least it is to me, that CAFs and not tumour cells emerge at the top of the network and that tumour cells are not observed as part of the network motif analysis. This is despite the fundamental requirement of tumour cells in establishing a TME (as it would be inflammation/fibrosis otherwise)? This is further substantiated by the observation that tumour cell conditioned medium bypass effects of CAFs on macrophage proliferation in vitro, which would be assumed to better represent the disease setting.

Response: Thank you for your endorsement and for your thoughtful comments. We agree, the finding that CAFs and not cancer cells are at the top is surprising. It may indicate that the breast cancer microenvironment mimics a fibrosis-like response, with cancer cells offering additional critical signals that can tune aspects such as the adaptive immune response and augment signals to macrophages. We provide more details in our response to the specific comments below.

Thus, 1) Does the network structure change between normal and tumour bearing tissue as analysed by scRNAseq (the dataset used also includes normal healthy tissue)? For example, are the motifs different with different cell composition?

Response: We thank the reviewer for this comment, which led us to add a set of new Supplementary Figure panels to the manuscript, describing the network and motif analysis of normal tissue (Figure S1C,F-G). We find that the network structure of normal breast tissue is different in an informative way. Fibroblasts are still near the top of the hierarchy, and macrophages are at the bottom (Figure S1F). Moreover, the strongest interaction of fibroblasts is still towards macrophages (Figure S1C). However, there are no significant 2-node motifs (Figure S1G). Instead the communication is more evenly spread among many cell types, as can be seen in the heatmap of interactions (Figure S1C).

In contrast, the cancer network has a more dominant hierarchy with a strong fibroblast autocrine loop, giving rise to the 2-node motif that we analyze. This strong fibroblast autocrine loop is also seen in fibrosis in the liver and in the heart (*Wang, S. et al. Sci. Transl. Med.*, 2023, *Miyara et al., bioRxiv* 2023), as mentioned in the Discussion.

We now added the following text to the results:

Lines 92-95: "*In normal breast tissue (from healthy individuals, see methods), the strongest interaction of fibroblasts is also with myeloid cells (comprised mainly of macrophages,*

hereafter termed macrophages), however both macrophages and fibroblasts engage in strong interactions with other cell types (Figure S1C).“

Lines 117-120: *“We also constructed a weighted and directed cell-network from the interaction matrix of the normal breast tissue. This analysis showed that fibroblasts are still near the top of the hierarchy and macrophages are at the bottom, however the network shows no significant 2-node motifs (Figure S1F-G).”*

We now added to the discussion:

Lines 527-530: *“The network for normal tissue showed no significant 2-node motifs. Motif analysis may thus help to discover important differences between normal and disease states, by revealing cell circuits that are crucial in each case.”*

Figure S1: The CAF-TAM subgraph is a dominant TME interaction. **A.** UMAP visualization shows the main markers for the human scRNA-seq¹⁴ clusters of fibroblasts (COL1A1), macrophages (CD68), and cancer (KRT18). **B.** Heatmap of interaction strengths between pairs of cells based on cumulative ligand-receptor interaction scores using CellChat¹⁵ applied to mouse breast cancer scRNA-seq data¹⁶. **C.** Heatmap of interaction strengths between pairs of cells based on cumulative ligand-receptor interaction scores using CellChat applied to normal breast tissue samples. **D.** The contribution of weight ranks to the total weights in the human breast TME network. 50% of the interactions contribute to 90% of the weights. **E.** four-cell circuits with different network motifs. In the human breast TME network. **F.** Illustration of the structure of the normal breast tissue network based on the analysis in (C) shows hierarchy. A weighted directed graph was generated from the interaction weights matrix. The network was pruned down by considering only the edges with the strongest 50% of weights. The network was then plotted using Layered Digraph Embedding to highlight the

inherent hierarchy. **G.** No statistically significant motifs were revealed by network motif analysis of 2-node circuits of the normal breast tissue network.

2) How is this reflected in the *in vitro* assay e.g. do the authors observe differences in their *in vitro* model +/- tumour cell conditioned medium that is reflected in the network analysis between healthy and tumour bearing tissue? This is of relevance in establishing whether the model reflects the *in vivo* condition, as hypothesised by the authors.

Response: The *in vitro* study is more similar to fibrosis than to normal tissue, even without cancer conditioned medium. We use the *in vitro* assay to explore perturbations to homeostasis. This includes regions where homeostasis is perturbed by high numbers of fibroblasts and circulating macrophages, which we denoted hot fibrosis, and regions with high numbers of fibroblasts but low numbers of macrophages which we denoted cold fibrosis. For this reason the macrophages used for this assay are bone marrow derived rather than tissue resident macrophages, mimicking a state of macrophage infiltration (as mentioned in the text, lines 190-193). *"We further asked whether addition of macrophages to an on-going culture of fibroblasts - simulating the infiltration of BMDMs into a tissue populated by resident fibroblasts - would yield similar interaction dynamics compared to those observed by simultaneous plating of both cell types."*

Moreover, the comparison between the network and structure with the *in vitro* data raise an interesting (and maybe important) question: Does the network structure/motifs or specific interactions/signals inform most critical elements of a normal vs diseased tissue?

Response: This insightful comment helped us add to the discussion the following statement:

Lines 527-530: *"The network for normal tissue showed no significant 2-node motifs. Motif analysis may thus help to discover important differences between normal and disease states, by revealing cell circuits that are crucial in each case."*

Specifically, tumour cell conditioned medium bypass macrophage dependence on fibroblasts but CAFs are at the top of the *in vivo* network structure (in Fig 1). Does this imply that specific tumour cell signals are critical and dominant over their position and connectivity within the network?

Response: We agree that tumor signals seem to bypass the macrophage dependence on fibroblasts. This may imply that tumor signals can be dominant over their position in the network.

We now add to the discussion (lines 559-561): *"This may imply that specific cancer cell signals are critical to shape the dynamics, despite the fact that cancer cells are located in the middle of the hierarchy in the *in-vivo* network."*

I remain unconvinced that use of tumour cell conditioned medium fully replicates the interactions observed in a more complex system e.g. conscripted CAFs and TAMs cannot feed back to the tumour cells and thus the system would be anticipated to only replicate a tumour to fibroblast/macrophage signalling axis. Could the authors either include data to clarify this point or simply acknowledge this possible limitation in the discussion.

Response: We thank the reviewer for the feedback. We have included a discussion of this limitation of our approach in the manuscript.

Lines 582-585: *"One limitation of our study is the use of cancer CM, which may not fully replicate the complexity of interactions observed. For instance, the experimental setup used to derive the cancer CM lacked fibroblasts and macrophages, and hence lacked the feedback from these cells to the cancer cells."*

Similarly, the authors denote fibroblasts as CAF and TAMs in their in vitro system, but in effect only replicate either normal or tumour-educated cells in vitro. The nomenclature should be clarified to avoid inconsistencies and misunderstandings.

Response: We have fixed the terminology in the revised manuscript.

Lines 394-400: *"To test whether CM induces CAF-like transcriptional signatures, we applied ssGSEA²⁸ on fibroblasts grown in control or cancer CM. ssGSEA analysis of fibroblasts grown in control medium demonstrated enrichment for both myCAF and iCAF signatures. (The apCAF signature was not detected in normal fibroblasts, consistent with our previously published notion that apCAFs are most likely not derived from tissue resident fibroblasts⁶). In the presence of cancer CM, however, the iCAF signature was no longer detected and the myCAF signature dominated the population (Figure 4G)."*

The authors have included a measurement of collagen deposition (Fig 2E) and are concluding the increase in production reflects increased fibroblast ECM production. However, isn't the collagen deposition simply a readout of increased numbers of fibroblasts rather than a measure of their function given the increase in fibroblast proliferation as macrophage numbers are increased?

Response: We thank the reviewer for this comment. We now clarify that the amount of collagen is normalized to the total amount of protein per well, so increased numbers of cells should not affect the normalized values. We added to the methods a sentence explaining that the results obtained are normalized to the total protein amount per well.

Lines 716-717: *"The collagen measurements obtained are normalized to the total protein amount per well."*

In Figure 4g the authors use ssGSEA to annotate CAF subsets, but I cannot see where they obtain the gene sets for the enrichment from – also, there is a lack of positive and negative controls for various CAF subtypes in this experiment.

Response: We now clarify these points. The gene sets used for the enrichment analysis were obtained from previously published studies from our lab (Halperin et al., *Cancer Res*, 2022; Hey et al., *Int. J. Cancer*, 2023). The CAF and TAM signatures are based on Bulk RNA-seq of TAMs and CAFs freshly isolated from 4T1 tumors in mice. These gene sets can be found in Supplementary Table 5. ssGSEA is an extension of GSEA designed to analyze the expression patterns of gene sets within individual samples rather than comparing between different conditions. It provides a quantitative score that represents the degree of enrichment of gene sets in a single sample (hence positive and negative controls are not required).

REVIEWERS' COMMENTS

Reviewer #4 (Remarks to the Author):

No further comments, the authors have addressed my concerns